# TypedThinker: Diversify Large Language Model Reasoning with Typed Thinking

**Danqing Wang**[1]*, **Jianxin Ma**[2], **Fei Fang**[1], **Lei Li**[1]
[1] Carnegie Mellon University     [2] Qwen Team
danqingw@cs.cmu.edu

## Abstract

Large Language Models (LLMs) have demonstrated strong reasoning capabilities in solving complex problems. However, current approaches primarily enhance reasoning through the elaboration of thoughts while neglecting the diversity of reasoning types. LLMs typically employ deductive reasoning, proceeding step-by-step from given conditions, which limits their exploration during problem-solving. Our analysis reveals that certain problems are exclusively solvable through specific reasoning strategies like inductive, abductive, or analogical reasoning. However, incorporating diverse reasoning approaches presents two key challenges: identifying the appropriate reasoning type for each problem and exploiting this approach during problem-solving. Therefore, we propose the `TypedThinker` that predicts suitable reasoning types based on the problem and their previous effectiveness and provides relevant demonstrations to guide LLMs in applying these strategies. Experimental results show significant improvements across multiple benchmarks, with performance gains of 3.4% for Mistral 7B, 6.5% for LLaMA3 8B, and 7% for Qwen 2 7B on logical and mathematical reasoning tasks. `TypedThinker` enhances LLM reasoning without requiring knowledge distillation from larger models. It can be integrated into more advanced systems like GPT-4o or specialized models like MetaMath to diversify their reasoning approaches and improve their problem-solving capabilities.

## 1 Introduction

Large Language Models (LLMs) exhibited promising capabilities in reasoning, such as solving logical reasoning and mathematical problems (Bai et al., 2022; OpenAI, 2023). Plenty of work has been done to improve the reasoning capabilities by adding reasoning thoughts (Wei et al., 2022) and making these thoughts more elaborated (Fu et al., 2023; Zheng et al., 2024). However, the exploration of diverse reasoning approaches remains severely limited. LLMs typically rely on a single reasoning pattern—usually deductive reasoning that proceeds step-by-step from given conditions. This narrow focus traps models in fixed thinking patterns, preventing them from solving problems that require different high-level reasoning approaches.

Current research fails to create truly diverse reasoning approaches. AlphaCode (Li et al., 2022; Leblond et al., 2023) tries to increase diversity by randomizing problem difficulty and tags. This method has limited scalability because it needs manual curation of attributes. It also works poorly beyond coding tasks. Increasing temperature settings offers another approach. This can generate outputs that appear different on the surface. Yet it rarely produces solutions with fundamentally different reasoning strategies. For example, repeated sampling (Brown et al., 2024) generated 100,000 solutions per problem using temperature 0.6. Their solutions [1] all follow the same basic approach. They start with problem conditions and work forward to deduce answers step-by-step. Humans, however, can use multiple reasoning strategies. One alternative approach is to propose a hypothesis first and then verify this hypothesis within the problem context. Current methods do not capture this diversity of thinking patterns.

---

*Work was done during the internship in Qwen Team.

[1]Their results are available at: https://huggingface.co/datasets/ScalingIntelligence/monkey_business.

Similarly, encouraging LLMs to explore diverse high-level thinking patterns leads to more varied solutions. Human cognitive research (Halpern, 2014; Bronkhorst et al., 2020) reveals multiple problem-solving approaches beyond deduction, including inductive (Flach and Kakas, 2000), abductive (Douven, 2011), and analogical reasoning (Bartha, 2013). These non-deductive strategies offer different directions through the solution space, often leading to more effective results. For example, abductive reasoning—proposing hypotheses and verifying them—is more suitable in multiple-choice scenarios. Humans demonstrate remarkable flexibility by switching from deductive to abductive reasoning when a free-response problem is reformatted with answer choices, despite unchanged problem content. This adaptive behavior shows how humans intuitively select optimal reasoning strategies for each problem format.

To examine how reasoning types affect LLM performance, we explicitly directed LLMs to apply specific reasoning strategies to problems. For each problem, the model is asked to follow a specific reasoning type and generate multiple solutions. We considered a problem "solvable" by a reasoning type if at least one of these solutions is correct. We calculate the percentage of problems solvable exclusively by one particular reasoning type. These represent cases where lacking the right reasoning approach makes problems nearly impossible to solve. We tested Mistral 7B instruct (Jiang et al., 2023) across four benchmarks: LogiQA (Liu et al., 2023a), BBH (Suzgun et al., 2022), GSM8k (Cobbe et al., 2021), and MATH (Hendrycks et al., 2021). Results in Figure 1 reveal that each reasoning type uniquely solves certain problems that other approaches cannot. This demonstrates that incorporating diverse reasoning strategies effectively expands the range of solvable problems. More detailed analysis is put in Section 4.2.

However, incorporating reasoning types into LLM problem-solving faces two major challenges. First, identifying the right reasoning type is difficult. Figure 1 shows that incorrect reasoning types mislead LLMs with wrong thinking directions. Second, ensuring the model follows the chosen reasoning type during problem-solving is crucial. Therefore, we propose `TypedThinker` to address these challenges. It predicts suitable reasoning types based on previous successful experiences and uses explicit demonstrations to help LLMs effectively apply them. `TypedThinker` consists of three key components: the **LLM reasoner**, the **meta-thinker** and **demonstration collection**.

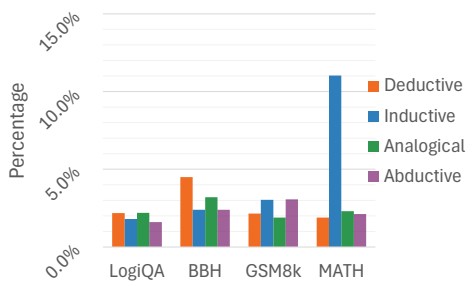

Figure 1: The percentage of problems solvable exclusively by one reasoning type.

The meta-thinker is fine-tuned on empirical effectiveness scores of each reasoning type from the training set. We use rejection sampling to collect successful solutions for each type as demonstrations, which enhance LLMs' ability to exploit specific reasoning strategies. During inference, `TypedThinker` uses the meta-thinker to identify the most suitable reasoning type and retrieves relevant demonstrations to guide the LLM reasoner in applying this approach to solve the problem.

Experimental results show that `TypedThinker` improves Mistral 7B instruct by 3.4%, LLaMA3 8B instruct (Touvron et al., 2023) by 6.5%, and Qwen 2 7B by 7% on two logical benchmarks and two mathematics benchmarks. We further demonstrate that `TypedThinker` can directly be applied to the benchmark Contexthub (Hua et al., 2024) and outperforms other baselines. Moreover, we show that the meta-thinker can be adapted in much larger LLMs such as GPT-4o or domain-specific LLMs such as MetaMath (Yu et al., 2023) and enhance their reasoning.

## 2 RELATED WORK

**Logical Reasoning** Logical reasoning includes various methods to emulate human-like thought processes (Wason and Johnson-Laird, 1972; Dowden, 2018; Nunes, 2012). Deductive reasoning focuses on deriving specific conclusions from general principles or premises, ensuring that conclusions logically follow if the premises are true (Johnson-Laird, 2010). In contrast, inductive reasoning involves generalizing from specific instances to broader principles, often used to identify patterns and make predictions based on empirical data (Flach and Kakas, 2000). Abductive reasoning, considered

more creative and open-ended, involves forming hypotheses to explain observations, often generating the most plausible explanation rather than a guaranteed conclusion (Douven, 2011). Analogical reasoning is concerned with the comparison between two or more objects and drawing a conclusion based on the similarity (Bartha, 2013). Previous LLMs studies on logical reasoning mainly focus on benchmarking its performance in different reasoning types (Bang et al., 2023; Dougrez-Lewis et al., 2024; Luo et al., 2023; Yu et al., 2024), or applying one reasoning type to solve the corresponding reasoning problems, such as using inductive reasoning for inductive reasoning problems (Wang et al., 2023a; Shao et al., 2024; Yang et al., 2024). Instead, this paper mainly focuses on the selection and application of the appropriate reasoning type when solving a general logic or math problem.

**Reasoning in Large Language Models** Plenty of studies have been done to enhance the reasoning capability of LLMs. Chain-of-thoughts methods focus on creating better instructions to improve the quality of the reasoning process, such as Complex CoT (Fu et al., 2023), Tree of Thought (ToT) (Yao et al., 2023) and Graph of Thought (Besta et al., 2024). Refinement-based methods revise LLMs solutions by the feedback from themselves or others model (Akyürek et al., 2023; Wang and Li, 2023). Search-based methods use the reward model to search the best reasoning path (Lightman et al.; Liu et al., 2023b; Hao et al., 2023). While most focus on creating high-quality reasoning paths, the diversity of thinking attracted more attention recently. Studies have investigated the diversity brought by repeated sampling (Brown et al., 2024) or multi-agent discussion with different prompts (Du et al., 2023; Liang et al., 2023; Suzgun and Kalai, 2024). Our paper aims to diversify thinking by incorporating suitable reasoning types for each instance.

**Self-improvement and Self-training in LLMs** Recent works explore the self-improvement capability of LLMs, by finetuning LLMs on their high-quality generations (Wang et al., 2023b; Huang et al., 2023; Toshniwal et al., 2024). This process can be extended to multiple iterations Gülçehre et al. (2023); Aksitov et al.. Benefiting the LLMs' ability to follow instructions, researchers also ask LLMs to provide feedback themselves and improve their responses without finetuning (Peng et al., 2023; Shinn et al., 2023). This can be further enhanced by using their own feedback as the reward model to provide better signals for finetuning (Yuan et al., 2024; Kumar et al., 2024). In this paper, we focus on stimulating their capabilities to conduct various reasoning types and use these experiences to diversify their thinking in reasoning type selection and following.

## 3 TYPEDTHINKER: DIVERSIFY THINKING WITH TYPED REASONING

In this paper, we focus on four logical reasoning types: deductive, inductive, abductive, and analogical reasoning defined in (Nunes, 2012). For each reasoning type, we provide a short definition and a simple example to demonstrate the inference rules, which are listed in Table 8 in the Appendix. Based on that, we introduce a reasoning framework `TypedThinker` to diversify LLMs' thinking with different reasoning types. As shown in Figure 2, there are three components in `TypedThinker`: the meta-thinker to select reasoning type, explicit collection for demonstration, and the LLM reasoner to exploit one particular reasoning type. `TypedThinker` optimizes the implicit policy of the meta-thinker and updates the explicit collection of demonstration based on previous experiences.

### 3.1 TYPING REASONING WITH IMPLICIT POLICY AND EXPLICIT DEMONSTRATION

Let $D = \{(\boldsymbol{x_1}, y_1), \cdots, (\boldsymbol{x_N}, y_N)\}$ be a set of $N$ problems, where $\boldsymbol{x}_i$ and $y_i$ is the problem and the ground-truth answer of the $i-$th instance. We define a reasoning type space $\mathcal{F}$ that includes an empty type and four types of reasoning: deductive, inductive, abductive, and analogical. The goal is to model the selection and implementation of various reasoning types as implicit and explicit collection of demonstration, thus enhancing LLMs' performance in reasoning tasks.

**Meta-thinker for the reasoning type identification.** Given a problem $\boldsymbol{x}$, the goal of the meta-thinker is to select an appropriate set of reasoning types to solve the problem. Specifically, it predicts an effectiveness score $s_k \in [0, 1]$ for each reasoning type $f_k \in \mathcal{F}$, which can be represented as $s_{x,k} = \pi_\theta(\boldsymbol{x}, f_k)$. $s_{x,k} = 0$ indicates that the problem $\boldsymbol{x}$ can hardly be solved by the reasoning type $f_k$ with a limited sampling times [2]. Note that the effectiveness scores of different types are independent of each other and their sums are not necessary to be 1. The most effective reasoning type is defined as $f^*(\boldsymbol{x}) = \arg\max_{f_k \in \mathcal{F}} s_{x,k}$. Meanwhile, we can obtain a set of reasoning types

---

[2]In this paper, we sample at most 10 times for one problem.

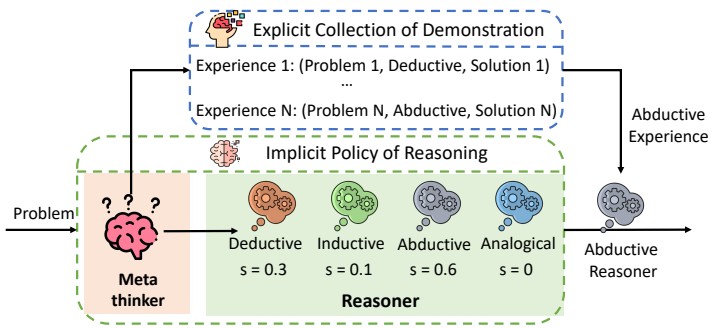

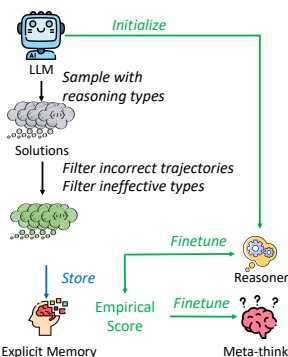

Figure 2: `TypedThinker` consists of three components: the meta-thinker to select the reasoning types, the explicit collection of demonstrations to retrieve relevant experience, and the reasoner to conduct the specific reasoning. The meta-thinker is fine-tuned to predict an effective score $s \in [0, 1]$ for each reasoning type.

Figure 3: Learning the implicit policy and explicit memory by self-training.

with a non-zero effective ratio, which we call the effective set: $F(\boldsymbol{x}) = \{f_k | s_{x,k} > 0\}$. We initialize $\pi_\theta$ with a pre-trained LLM. The prompt is listed in Appendix A.1.

**Explicit collection of demonstration** `TypedThinker` collects a set of demonstration $M = \bigcup_{f_k \in \mathcal{F}} M_k$ for each type of reasoning. For each problem in the training set, we keep one correct solution per reasoning type if applicable, resulting in a set of at most $|D| \times |\mathcal{F}|$ solutions. If multiple solutions exist for one problem, we keep the longest to get a more detailed context. The entry in the collection of demonstration $\mathcal{M}_k$ is represented as a tuple of $(\boldsymbol{x}_r^k, sol_r^k)$. Here $sol_r^k$ is the concrete reasoning process of the reasoning type $f_k$, including the predicted answer. During inference stage, given a new problem $\boldsymbol{x}$ and its reasoning type $f_k$, it retrieves a set of relevant experience $d_x^k = \{(\boldsymbol{x}_r^k, sol_r^k) \in M_k | L(\boldsymbol{x}_r^k, \boldsymbol{x}) < \delta\}$. $L$ is the distance function measuring relevancy between two problems and $\delta \in [0, 1]$ is the relevancy threshold. We use the cosine similarity between the semantic embeddings as the distance function. The retrieved experiences are used as the few-shot examples of the reasoner.

**Reasoner to perform the reasoning according to the type.** The reasoner applies the reasoning type $f_k$ to the problem $\boldsymbol{x}$ and provides a detailed reasoning path for its predicted answer $\hat{y}$. The reasoner is based on LLM to conduct reasoning and the instruction is composed of $(\boldsymbol{x}, f_k, d^k)$, where the $d^k$ is the retrieved relevant successful experience. The reasoner can be further optimized via instruction tuning to enhance the capability of conducting a specific type of reasoning.

To conclude, `TypedThinker` enhances LLMs' reasoning by estimating the effectiveness of each reasoning type and guiding the LLM reasoner with the demonstration of this reasoning type. Specifically, the meta-thinker $\pi_\theta$ predicts an effective score $s_k$ for each reasoning type. `TypedThinker` then retrieves the most relevant reasoning demonstration $d^k$ from the fixed explicit collection corresponding to the reasoning type $f_k$. Finally, the reasoners conduct the specific type of reasoning $f_k$ with the guidance of demonstration. In our experiments, we use two approaches to decide the reasoning types used in the problem-solving: One is to greedily resample several times based on the most effective reasoning type $f^*$ and use self-consistency (Wang et al., 2022) to enhance the answer; the other is to sample solutions for all effective reasoning types $F$, and apply a weighted vote with the effective score as the coefficient. By default, we use the greedy approach for `TypedThinker`, and we discuss the weighted vote in Section 4.4.

## 3.2 OPTIMIZE IMPLICIT POLICY FOR REASONING TYPE SELECTION AND EXPLOITING

We optimize the meta-thinker $\pi_\theta$ and the reasoner while updating the explicit collection of demonstration with the collected experience. The pipeline is demonstrated in Figure 3. The green lines represent the parametric optimization process, while the blue line represents the non-parameter update.

**Diversify Reasoning Experiences with Types** To inspire LLMs' knowledge of solving problems with different reasoning types, the definition (Table 8) and manually-written few-shot examples with detailed reasoning paths (Table 16) are used for prompting solutions for each reasoning type. For

each problem in the training set, we use a temperature of 1 to sample 10 solutions per reasoning type. These solutions are then filtered by the correctness of the final answers. To guarantee that these solutions belong to their reasoning type, we apply a *reverse check* on the remaining solutions. For the experience $(\boldsymbol{x}, sol, y)$ of the reasoning type $f_k$, we prompt the model to predict its reasoning type $\hat{f}_k$. If $f_k = \hat{f}_k$, we think this experience indicates the methodology of this reasoning type and keep it. Otherwise, it will be removed. Finally, we collect an experience dataset $\overline{D}$ with multiple types of reasoning. The experiences are grouped by their reasoning type and are stored in the explicit collection of demonstration $M$.

**Optimize the Implicit policy of Meta-thinker and Reasoner** Given a problem $\boldsymbol{x}$, the meta-thinker $\pi_\theta$ predicts a score $s_{x,k}$ to indicate how likely this reasoning type can solve this problem. This can be estimated by the experience in the training set. We assume that if one reasoning type is more effective in solving this problem, it will generate more correct solutions among the same sampling times. Therefore, given there are $n_k$ successful experiences of the reasoning type $f_k$ among $m$ samples, we define the empirical effectiveness score based on its success rate: $\overline{s}_{x,k} = n_k/m$. This empirical effectiveness score calculated on the experience dataset $\overline{D}$ is then used for finetuning the meta-thinker. We reconstruct the tuple $(\boldsymbol{x}, f_k, \overline{s}_{x,k})$ into the instruction-following pair via the prompt in Section A.1 for supervised finetuning. Meanwhile, we finetune a reasoner with the experience to enhance its capability to conduct a specific type of reasoning.

# 4 EXPERIMENTS

## 4.1 EXPERIMENT SETUP

We investigate two open-source LLMs Mistral 7B instruct (Jiang et al., 2023) and LLaMA3 8B instruct (Touvron et al., 2023) and Qwen 2-7B-Instruct (Bai et al., 2023) on two logical benchmarks (LogiQA, BBH) and two mathematics benchmarks (GSM8K and MATH). For each LLM, we set up the following baselines: (i) **Few-shot baseline** with 3 in-context examples. We use the few-shot examples provided in Suzgun et al. (2022) for BBH, and text-based few-shot examples in Toshniwal et al. (2024) for GSM8k and MATH since we do not consider the code interpreter in this paper. We also manually write few-shot examples for LogiQA, (ii) **CoT Selection**: Select the best reasoning type by prompting. We let the LLM identify the best reasoning type and then apply the selected type to the problem. (iii) **Self-Discover** (Zhou et al., 2024) generates a task-level reasoning structure by prompting LLMs to select relevant modules from a list of seed modules and adapt the selected module to task-specific descriptions. We follow their official implementation [3] and use the backbone LLMs to generate one reasoning structure from an exemplar training instance of each task. This reasoning structure is then applied to all instances in this task. (iv) **Zero-shot Mixture of Reasoning (MoR)**: apply all possible reasoning types and use the majority vote to get the final answer [4]. The LLM is instructed with the definition and demonstration in Table 8. (v) **Few-shot MoR**: Similar to the zero-shot MoR except for each reasoning type, 3 few-shot examples are provided in the prompt. (vi) `TypedThinker`: use the most effective reasoning type $f^*$. The +SC baselines indicate the majority vote over 5 responses.

The temperature is set to 0.7 for all baselines as suggested by Wang et al. (2022). The maximum output length is set to 1000 tokens. We use SentenceTransformer[5](Reimers and Gurevych, 2019) to retrieve top-3 similar experiences and the threshold is set to $\delta = 0.5$. We use accuracy as the measurement of task performance. The model response is compared with the ground truth based on the exact match for logic problems. The script in Toshniwal et al. (2024) calculates the mathematical equivalent of mathematics benchmarks. The training details are put in Appendix A.3.

## 4.2 HOW DO REASONING TYPES ENCOURAGE DIVERGENT THINKING DURING GENERATION?

We first investigate the role of reasoning types in LLMs' self-training. We group problems of the collected experience dataset $\overline{D}$ based on their empirical effective set $\overline{F}(\boldsymbol{x}) = \{f_k | \overline{s}_{x,k} > 0\}$, and the empirical effectiveness score is defined in Section 3.2. The problems in the same group can be solved

---

[3]https://github.com/kailashsp/SELF-DISCOVER
[4]For answers with the same votes, we choose the first one in alphabetical order.
[5]https://www.sbert.net/

with the same set of reasoning types. We focus on the effective set with only one reasoning type and count the size of these groups. The size indicates how many problems that can only be solved by one specific reasoning type, showing the advantage of including this reasoning type. We illustrate their percentage on the whole dataset in Figure 1. We find that even if we use temperature = 1 to sample 10 times to diversify the solutions, a lot of problems still have only one effective reasoning type. *It indicates that given an inappropriate reasoning type, the diversity brought by repeated sampling with a high temperature cannot help the LLM solve this problem.*

Meanwhile, although these reasoning types have similar performance on the whole dataset (shown in Table 6 in Appendix), the problems they can solve do not completely overlap. None of the percentages of the reasoning type is zero, indicating that for each reasoning type, there is a unique set of problems that can only solved by it. It indicates that these reasoning types have their advantages over different problems, highlighting the importance of considering the appropriate reasoning types during problem-solving.

We further compare the diversity of the solutions before and after adding the reasoning types in Table 10 of Appendix A.4. We can find that introducing different reasoning types can bring more diversity to the solution set than repeated sampling with a high temperature.

Table 1: `TypedThinker` achieves the best performance in both single response and the majority vote setting on two logical benchmarks and two math benchmarks. @5 indicates the result is based on the majority vote over 5 responses. +SC indicates the self-consistency method. MoR indicates the Mixture of Reasoning, which employs all reasoning types (including an empty type) and votes for the final output. Avg. indicates the average accuracy over four benchmarks. Qwen's results are put in Table 11.

| | Mistral 7B | | | | | LLaMA3 8B | | | | |
|---|---|---|---|---|---|---|---|---|---|---|
| | LogiQA | BBH | GSM8K | MATH | Avg. | LogiQA | BBH | GSM8K | MATH | Avg. |
| Few-shot | 0.485 | 0.346 | 0.369 | 0.074 | 0.318 | 0.581 | 0.359 | 0.581 | 0.193 | 0.428 |
| + SC @5 | 0.532 | 0.441 | 0.444 | 0.136 | 0.388 | 0.579 | 0.391 | 0.769 | 0.250 | 0.497 |
| CoT Selection | 0.474 | 0.361 | 0.372 | 0.095 | 0.325 | 0.564 | 0.392 | 0.556 | 0.181 | 0.423 |
| + SC @5 | 0.503 | 0.429 | 0.466 | 0.132 | 0.382 | 0.562 | 0.426 | 0.785 | 0.222 | 0.499 |
| Self-Discover | 0.386 | 0.340 | 0.141 | 0.056 | 0.231 | 0.493 | 0.425 | 0.587 | 0.200 | 0.426 |
| + SC @ 5 | 0.476 | 0.391 | 0.208 | 0.082 | 0.289 | 0.540 | 0.543 | 0.701 | 0.278 | 0.516 |
| Zero-shot MoR @5 | 0.528 | 0.414 | 0.313 | 0.108 | 0.341 | 0.556 | 0.463 | 0.666 | 0.189 | 0.468 |
| Few-shot MoR @5 | 0.509 | 0.456 | 0.460 | 0.127 | 0.388 | 0.599 | 0.543 | 0.585 | 0.195 | 0.481 |
| `TypedThinker` | 0.554 | 0.423 | 0.386 | 0.092 | 0.364 | 0.599 | 0.543 | 0.585 | 0.195 | 0.481 |
| + SC @5 | 0.570 | 0.469 | 0.500 | 0.149 | **0.422** | 0.637 | 0.591 | 0.753 | 0.267 | **0.562** |

### 4.3 WHAT KINDS OF BENEFITS CAN TYPEDTHINKER BRING?

As we can see in Table 1, our `TypedThinker` achieves the best performance among baselines. The improvement is more obvious in LLaMA3 8B, which is more powerful than Mistral 7B. It shows that LLMs with a better capability in reasoning and instruction-following can benefit more from the self-training of `TypedThinker`. Additionally, there are several key insights from the detailed comparison with different baselines.

**Appropriate reasoning types improve the reasoning performance.** The main difference between Fewshot and CoT Selection without the majority vote is the reasoning type selection. For CoT Selection, the model is first prompted to predict a reasoning type and then apply it, while the Fewshot baseline directly solves the problem. However, we find that the CoT Selection struggles with the reasoning type selection. Given the option to choose from four reasoning types or none, it chooses none over 60% of the time. The rest of the time, it selects more than 50% deductive, while only 34% of them can be effectively solved by deductive reasoning during the sampling. The mismatch in reasoning types results in poor performance. Facilitating with a trained meta-thinker, `TypedThinker` is more accurate in selecting the reasoning type, which helps it improve performance under the single response setting. Self-Discover, which uses a shared reasoning structure for all instances of the task, performs poorly, especially for the weaker model Mistral 7B. This may be due to the difficulty these models face in identifying a reusable shared high-level reasoning structure for diverse instances, especially in datasets like LogiQA, where reasoning structures are highly varied.

**Precise prediction is more effective than an inappropriate mixture.** The zero-shot MoR and few-shot MoR apply all types of reasoning to the given problem and use a majority vote to get the final answer. Compared with the other two majority vote baselines Few-shot + SC @5 and CoT Selection + SC @5, these methods fall behind on several benchmarks, especially on MATH. We find that the performance drop typically happens when there are only one or two reasoning types that are effective for this problem. In such cases, the majority of incorrect answers dominate, resulting in fewer votes for the correct one. As we can see in Figure 1, plenty of problems on the MATH benchmark can only be solved by inductive reasoning. In such cases, if the CoT Selection correctly predicts the inductive reasoning for them, the CoT Selection + SC @5 can benefit from the majority vote and have a better performance. This highlights the importance of predicting the effectiveness of reasoning types before aggregating them.

**Experience of how to conduct a specific type of reasoning is important.** The performance difference between zero-shot and few-shot MoR illustrates the impact of the reasoning demonstration. When prompted solely with the definition, LLMs struggle to understand how to apply the reasoning type to specific problems. It can be improved by human-written few-shot examples in few-shot MoR. However, it still falls behind the non-parametric retrieval and the parametric reasoner in `TypedThinker`, both of which enhance the capability of conducting a specific reasoning type. Additionally, poor performance in Self-Discover also indicates that, without demonstration, the complex reasoning structures will introduce excessive complexity, confusing the models.

Table 2: Ablation Study on the Mistral 7B based `TypedThinker`'s components. We remove one component each time. The results are based on the best reasoning type and calculated for the single response per query. The negative scores indicate the performance drop, and the largest scores are shown in bold.

|  | LogiQA | BBH | GSM8K | MATH |
|---|---|---|---|---|
| TypedThinker | 0.554 | 0.423 | 0.386 | 0.092 |
| w/o Finetuned Reasoner | -0.076 | -0.041 | -0.102 | -0.018 |
| w/o Meta-thinker | -0.025 | -0.036 | **-0.152** | **-0.024** |
| w/o Memory | **-0.082** | **-0.051** | -0.033 | 0.013 |

## 4.4 WHAT CONTRIBUTES TO TYPEDTHINKER'S EFFECTIVENESS?

We conduct several investigations to enhance the understanding of our proposed method.

**Ablation study** The ablation studies are conducted on three key components in `TypedThinker`. Each time one component is removed. It includes (i) **w/o Fine-tuned Reasoner**: it is replaced with the base LLM (ii) **w/o Meta-thinker**: it is replaced with a CoT selection (iii) **w/o collection of demonstration**: the explicit collection is replaced with the human-written few-shot examples of each reasoning type. In Table 2, we can find the meta-thinker is the most important module for the math benchmarks, while the explicit collection is more effective on two logical benchmarks. The fine-tuned reasoner also contributes a lot to the performance improvement. We also observe that explicit collection does not always bring benefits: the performance on MATH even increases when we remove it. We find that the retrieved examples usually have a similar context but different numbers. The math calculation in the retrieved chain-of-thoughts solutions will mislead the reasoner. This is consistent with the observations of Toshniwal et al. (2024) that the solutions with masked computations are more beneficial to the math problems. For logical problems, there are fewer calculations and the retrieved solutions focus more on the reasoning process.

**Meta-thinker's predictions achieve a high correlation with the empirical effectiveness score.** We evaluate the performance of the meta-thinker by the correlation between the predicted effectiveness score and the empirical one (which we view as the ground truth). We split the collected experience dataset $\overline{D}$ by problems and use 0.9 of them to train the meta-thinker and 0.1 for testing. We use Kendall's $\tau$ coefficient to evaluate the correlation. It measures rank correlation, essentially assessing the similarity of orderings when data is ranked. A higher Kendall's $\tau$ coefficient indicates that when the ground truth assigns a high effectiveness score to a reasoning type, the meta-thinker also ranks it high, thereby validating the reliability of the predicted scores. We compare the performance under three settings: the meta-thinker trained only on the logical domain, only in the math domain, and jointly trained on the unified domains (including both logic and math data). The meta-thinker trained

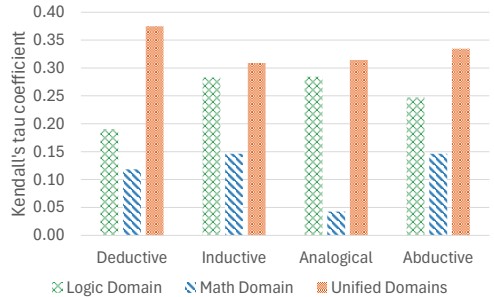 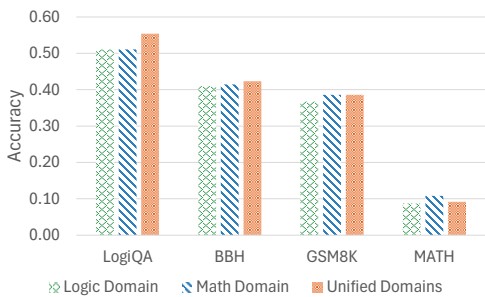

Figure 4: Kendall's $\tau$ coefficient between the prediction confidence score with the ground truth. All results have the p-value $< 0.05$. The unified policy shows the best correlation on all reasoning types.

Figure 5: Task performance of `TypedThinker` with meta-thinkers trained on different domains. The unified one performs best in most cases except the MATH benchmark, where the pure math setting dominates.

on the unified domain achieves the highest correlation. This suggests that training on a dataset with multiple domains enhances the meta-thinker's ability to accurately rank and predict suitable reasoning types, thereby improving its overall performance. We also calculate the accuracy between the predicted optimal reasoning type and the empirical one for the unified setting. The average accuracy on four benchmarks is 68.3% (LogiQA 75.4%, BBH 75.6%, GSM8k 72.1%, and MATH 47.7%). Note that the meta-thinker can predict an incorrect optimal reasoning type $f_k$ while still generating a correct solution. It is because the predicted reasoning type can belong to the effective set $F = \{f_k | s_{x,k} > 0\}$, indicating the reasoning type can also help solve the problem.

**Unified meta-thinkers perform well in most cases.** We further investigate the effectiveness of these policies by facilitating `TypedThinker` with these meta-thinkers. The results are based on the Mistral 7B `TypedThinker` without SC. The results in Figure 5 show that the unified meta-thinker has the best performance in most cases. However, in the more difficult MATH dataset, the specific meta-thinker trained in the math domain can help it be more powerful. To conclude, the unified meta-thinker has reasonable performance in all its domains, while for difficult problems it may slightly underperform the specific meta-thinker trained in this domain.

Table 3: `TypedThinker`'s performance with the most effective reasoning type $f^*$ v.s. weighted votes on the effective set $F$.

Table 4: `TypedThinker` performs best on the unseen benchmark Contexthub. Here the results are based on the majority vote over 5 responses (+SC @5).

|  | LogicQA | BBH | GSM8K | Math | Average |
|---|---|---|---|---|---|
| **Mistral 7B** | | | | | |
| SC @5 on $f^*$ | 0.570 | **0.469** | **0.500** | **0.149** | **0.422** |
| weighted on $F$ | **0.581** | 0.453 | 0.501 | 0.127 | 0.416 |
| **LLaMA3 8B** | | | | | |
| SC @5 on $f^*$ | **0.637** | **0.591** | **0.753** | **0.267** | **0.562** |
| weighted on $F$ | 0.619 | 0.587 | 0.738 | 0.245 | 0.547 |

|  | Mistral 7B | LLaMA3 8B |
|---|---|---|
| Few-shot | 0.419 | 0.378 |
| CoT Selection | 0.415 | 0.390 |
| Zero-shot MoR | 0.415 | 0.403 |
| Few-shot MoR | 0.432 | 0.390 |
| Self-Discover | 0.332 | 0.365 |
| `TypedThinker` | **0.452** | **0.423** |

**Optimal reasoning type v.s. weighted vote on the effective set** In the main experiment, we use the optimal reasoning type $f^*$ which has the highest effectiveness score for reasoning. As discussed in Section 3.1, we can also use a majority vote on the effective set $F$ with the effectiveness score as the coefficient. Specifically, if one solution is based on a reasoning type with a higher effectiveness score, its vote gets a larger weight. The results are shown in Table 3. We can see that the weighted vote can balance different reasoning types on LogiQA and GSM8k for the Mistral-7B-based model. However, on the other two benchmarks, the `TypedThinker` + SC @5 has a better performance. It indicates that accurate selection is more important if one or two reasoning types dominate the benchmark. For example, as we have shown in Figure 1, there are a lot of problems that can only be solved by inductive reasoning, indicating the other types will mislead the final answer. In such cases, the self-consistency of inductive reasoning is more powerful than the weighted vote. However, when we have a more accurate meta-thinker that can identify the correct reasoning type and a more

powerful reasoner that can follow the specific reasoning type, for example, models initialized by LLaMA3, the advantage of `TypedThinker` + SC is more obvious.

## 4.5 CAN TYPEDTHINKER BE APPLIED TO NEW DOMAINS OR NEW LLMS WITHOUT FINETUNING?

It is essential to evaluate the generalization capability of `TypedThinker`. We assess it from two aspects: (i) `TypedThinker`'s performance on new domains; and (ii) other LLMs' performance after facilitated with our finetuned meta-thinker and the explicit collection of demonstration.

**`TypedThinker` generalizes well to the unseen domain.** We use a new propositional logic benchmark Contexthub (Hua et al., 2024) for evaluation. It is a recently released dataset, which has never been seen by Mistral and LLaMA3 models during the pre-training. It contains problems from 12 categories with 4 levels of difficulty. We select the difficulty of level 4 to test the complex logic reasoning capabilities. We use the experiences collected from LogiQA as the explicit collection of demonstration. The meta-thinker and the reasoner are fine-tuned on four training benchmarks. The results in Table 4 show that `TypedThinker` outperforms other baselines on this unseen domain as well, indicating that it can generalize well to new domains. One interesting thing is Mistral 7B baselines significantly outperform LLaMA3 8B on this benchmark and its superior capabilities make it benefit more from our `TypedThinker`.

**Facilitating LLMs with `TypedThinker` makes them more powerful.** Our `TypedThinker` framework is orthogonal to the backbone LLMs and can be adapted to new LLMs. There are two ways to use a new LLM in the `TypedThinker` framework: one is to conduct the self-training process, like the two LLMs used in our main experiments (Mistral 7B and LLaMA3 8B); the other is to use our fine-tuned meta-thinker for reasoning type selection and the explicit collection of demonstration for retrieval while using the new LLM as the reasoner without finetuning. The first way can make the LLM more powerful (as shown in the performance comparison between `TypedThinker` and `TypedThinker` w/o Finetuned Reasoner in Table 2), but the latter one is more flexible. Here we use the second way to evaluate the direct transferability to new LLMs. We choose one of the most powerful LLMs GPT-4o and one math-specific 7B model MetaMath (Yu et al., 2023). MetaMath is a Mistral-7B-based model trained with more than 400k synthesized math data distilled from GPT-3.5-Turbo. We randomly sample 100 examples from each benchmark for GPT-4o. For MetaMath, we use the whole test set. The results are shown in Table 5 and Table 6. Compared with Mistral 7B in Table 1, the high-quality and large scale of synthesized data from GPT-3.5-Turbo enhances MetaMath's capabilities in math problems. `TypedThinker` can further improve its performance by reasoning type selection and explicit collection. Meanwhile, although the superior performance of GPT-4o on two math datasets leaves little space for improvement, the results on logic benchmarks (LogiQA and BBH) demonstrate that the meta-thinker trained with the small 7B model also enhances its performance. These findings confirm that our approach is not only effective in improving smaller LLMs but also transferable to larger models, further validating the generalization capability of `TypedThinker`.

Table 5: GPT-4o's performance is improved with our meta-reasoner. We use the finetuned Mistral 7B meta-thinker to predict the reasoning type.

| | LogiQA | BBH | GSM8k | MATH |
|---|---|---|---|---|
| GPT-4o | 0.76 | 0.84 | 0.97 | 0.89 |
| + SC @ 5 | 0.80 | 0.85 | **0.98** | 0.90 |
| TypedThinker | 0.80 | 0.86 | 0.95 | 0.88 |
| + SC @5 | **0.81** | **0.90** | 0.96 | **0.91** |

Table 6: `TypedThinker` can also enhance the performance of the math-specific 7B model such as MetaMath.

| | GSM8k | MATH |
|---|---|---|
| MetaMath | 0.690 | 0.209 |
| + SC @ 5 | 0.704 | 0.220 |
| TypedThinker | 0.696 | 0.220 |
| + SC @ 5 | **0.736** | **0.246** |

## 4.6 CASE STUDY

Here is one example of `TypedThinker` on the LogiQA benchmark in Table 7. This problem states a phenomenon that a higher altitude leads to a lower atmospheric pressure. Based on this observation, it is easy for humans to use inductive reasoning and get a general conclusion about the

inverse cause-and-effect relationship. It is also natural for humans to use analogical reasoning and find the most similar options. The meta-thinker gives the highest effectiveness score for inductive reasoning, which is then chosen as the optimal reasoning type $f^* =$ inductive. Effectiveness scores are all larger than 0, so the effective set is all reasoning types. The reasoner gets the correct answer for deductive and inductive reasoning while doing wrong on the other reasoning types. If we use the majority vote over five answers, (A) and (C) will have the same votes, indicating that there is a 50% chance to be correct [6]. However, with the effectiveness score predicted by the meta-thinker, `TypedThinker` can get the correct answer either by applying the optimal reasoning type or using the weighted vote on the four answers. Besides, without a specific reasoning type (which is 'Empty'), the model cannot arrive at the correct answer. This shows the limitation of the common few-shot baselines. It shows that `TypedThinker` improves the reasoning performance by the introduction of diverse reasoning types and the capability of selecting the appropriate type to apply.

Table 7: One example from LogiQA. The correct answer and the reasoning type with the highest effectiveness score are underlined. **MoR** is the few-shot MoR baselines, which use the majority votes among reasoning types.

| | |
|---|---|
| Problem | **The higher the altitude, the smaller the atmospheric pressure. Because the altitude of Place A is higher than that of Place B, the atmospheric pressure of Place A is lower than that of Place B. Which of the following examples shows the same pattern of reasoning?** 
 *(A) In a highly competitive market, the better the product quality and the more advertising investment, the greater the product sales. Company A invests more money in advertising than Company B. So Company A sells more products than Company B.* 
 *(B) The older a person is, the more mature he becomes. Person A is older than his son, so Person B is more mature than his son.* 
 *(C) The older a tree is, the more rings it has. The age of the locust tree in A's yard is older than that of B's family, so the locust tree of A's family has more rings than B's.* 
 *(D) The greater the vocabulary of a language, the more difficult it is to learn. English is harder to learn than Italian, so English has a larger vocabulary than Italian.* |
| Ground Truth | (C) |
| Predicted scores and their answers | Deductive: 0.4; Inductive: 0.5; Analogical: 0.4; Abductive: 0.4; Empty: 0.4 
 Deductive: (C); Inductive: (C); Analogical:(A); Abductive: NULL; Empty: (A) |
| Model Output | **MoR**: (A); **TypedThinker with** $f^*$ (C); **TypedThinker with** $F$: (C) |

## 5 CONCLUSION AND LIMITATION

We investigate how reasoning types diversify LLMs' thinking and propose `TypedThinker` to incorporate different reasoning types into problem-solving. `TypedThinker` is inspired by human cognition processes during reasoning: it learns an implicit policy to select the appropriate reasoning types with the meta-thinker and to apply the selected type of reasoning with the reasoner. It also maintains an explicit memory to retrieve experiences to aid reasoning. The results show that `TypedThinker` enhances the reasoning capabilities of Mistral 7B, LLaMA3 8B and Qwen 2 7B on four benchmarks. Furthermore, `TypedThinker` shows good generalization capabilities in new domains and models.

Despite the promising results, `TypedThinker` has several limitations that need further investigation. Firstly, one problem may require different reasoning types at different steps, and applying one sole reasoning type can hardly find a correct solution. In that case, dividing the problems into multiple reasoning steps, and applying `TypedThinker` for each step could make the reasoning more diverse and effective. Additionally, this paper mainly focuses on logical and mathematical benchmarks. Expanding to a broader range of tasks, such as code generation and creative problem-solving, could deepen the understanding of the role of reasoning types in various problems and provide a more comprehensive assessment of `TypedThinker`'s capabilities.

---

[6]In our implementation, answers with the same votes are ranked based on their alphabetical order, so (A) will be chosen in this case.

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

# A    APPENDIX

## A.1    PROMPT

We introduce the simple and practical definition of four reasoning types in Table 8. Table 16 lists the few-shot examples of each reasoning type. The full few-shot examples can be found in the supplementary materials. We use the same few-shot examples for the logical problems and create another set of examples for the mathematics problems.

Table 8: Description of different reasoning types. We give informal definitions that are easy to follow and illustrate simple examples for each reasoning type.

| Type | Definition | Example |
|------|-----------|---------|
| **Deduction** | *Deduce conclusion based on the general rules and premise.* | From the premises 'all frogs are amphibians' and 'no cats are amphibians', we can infer the conclusion 'no cats are frogs' |
| **Induction** | *Make broad generalizations from specific observations.* | Starting from the empirical observation that 'all ravens I have seen so far are black', inductive reasoning can be used to infer that 'all ravens are black' |
| **Abduction** | *Assume one candidate is correct and check whether it meets the condition in the problem.* | Guess that it has rained to explain that the streets are wet. A tsunami could also explain why the streets are wet but this is usually not the best explanation. |
| **Analogy** | *Retrieve several relevant information and draw the conclusion of this problem based on the similarity.* | Infer information about humans from medical experiments on animals: (1) rats are similar to humans; (2) birth control pills affect the brain development of rats; (3) therefore they may also affect the brain development of humans. |

The prompt used by the meta-thinker is:

> *Given the question below, please identify the type of reasoning required to provide a solution. You may choose the following reasoning types: Deductive, Inductive, Analogical, Abductive Reasoning, or None. None indicates that no specific reasoning type is needed for this problem. Please assign an effectiveness score for each reasoning type from 0 to 1, where 0 represents no effective and 1 represents full effective. Please return the reasoning types and their corresponding effectiveness scores in the JSON format.*
>
> *For instance, if you think the question can be solved using both deductive and inductive reasoning, with an effectiveness of 0.5 for deductive reasoning and 0.3 for inductive reasoning, you should return: [{"ReasoningType": "Deductive", "Effectiveness": 0.5},{"ReasoningType": "Inductive", "Effectiveness": 0.3},{"ReasoningType": "Analogical", "Effectiveness": 0},{"ReasoningType": "Abductive", "Effectiveness": 0}, {"ReasoningType": "None", "Effectiveness": 0}].*

The prompt used by the reasoner is listed below. The definition is based on Table 8.

> *Use [$f_k$] reasoning to solve the given question. [$f_k$] reasoning is [definition].*

## A.2    DATASET

### A.2.1    DATA PROCESSING

The dataset statistics of the four benchmarks are detailed in Table 9. For multiple-choice questions, we calculate accuracy using the exact match criterion. For mathematics problems, we compare the model's response with the ground truth using mathematical equality.

LogiQA (Liu et al., 2021; 2023a) is a multi-choice understanding benchmark for logical reasoning. It follows the definition of DeLancey (2017) and categorizes the problems into categorical reasoning,

Table 9: Logical benchmarks and mathematic benchmarks we used in this paper. We follow the standard train/test split on LogiQA and follow the split in Toshniwal et al. (2024) for GSM8k and MATH. For BBH, we randomly split the dataset. The synthesized data is described in Section 3.1. BBH includes 16 tasks while MATH includes math problems of 7 categories. Policy indicates the data used to train the meta-reasoner and SFT indicates the instruction-following in reasoner finetuning.

| | | Benchmark | | | | Empirical Dataset | |
|---|---|---|---|---|---|---|---|
| | # Task | # Train | # Val | # Test | # Total | # Meta-thinker | # Reasoner |
| LogiQA | 1 | 3757 | 500 | 511 | 4768 | ~2k | ~6k |
| BBH | 16 | 1904 | 320 | 1600 | 3824 | ~1k | ~3.5k |
| GSM8k | 1 | 6473 | 1000 | 1319 | 8792 | ~4k | ~4k |
| Math | 7 | 6500 | 1000 | 5000 | 12500 | ~1k | ~1k |

sufficient conditional reasoning, necessary conditional reasoning, disjunctive reasoning, and conjunctive reasoning. These reasoning categories are not orthogonal and one problem can belong to multiple categories. We follow the standard training/validation split and only keep examples with more than 3 reasoning categories. This makes the problem more diverse and difficult to solve. We take the validation set as the test set and randomly select 500 examples from the training set for validation.

BBH (Suzgun et al., 2022) is a set of hard problems borrowed from Big Bench (Srivastava et al., 2022). They are also formatted as multi-choice problems. We pick the English tasks with more than 2 options, resulting in 16 tasks: date understanding disambiguation qa, geometric shapes, hyperbaton, logical deduction three, logical deduction five, logical deduction seven, movie recommendation, penguins in a table, reasoning color, ruin names, snarks, temporal sequences, tracking shuffled three, tracking shuffled five, and tracking shuffled seven. For each task, we randomly select 100 examples as the test set and 20 examples as the validation. The rest are used as training examples.

GSM8k (Cobbe et al., 2021) is a commonly used math benchmark to evaluate LLMs' capability in math reasoning. It contains 8.5K grade school math word problems, which are split into 7.5k training examples and 1k test problems. Each problem usually takes between 2 and 8 steps to solve. MATH (Hendrycks et al., 2021) is also a popular math benchmark for LLMs. It contains 12,500 challenging competition mathematics problems with 7 categories. There are 7.5k training examples and 5k test problems. We follow Toshniwal et al. (2024) to process the dataset.

Contexthub (Hua et al., 2024) is a new propositional logic benchmark. It contains abstract and contextualized logical problems from 12 categories with 4 levels of difficulty (Zhu et al., 2024). We follow the standard split of the original paper and use the subset of difficult level 4 to test the complex logic reasoning capabilities. The abstract logical problems only contain the symbolic variable without natural language description, which can be viewed as symbolic reasoning problems.

Livebench (White et al., 2024) is a recently proposed benchmark with 18 diverse tasks across 6 categories, specifically designed to minimize data contamination. All problems have verifiable, objective ground-truth answers, allowing hard questions to be scored accurately and automatically. We evaluate our models on three tasks (*spatial, web of lies v2, zebra puzzle*) from the reasoning category, splitting them 0.7/0.3 for training and testing.

### A.2.2 DATASET EXAMPLES

We demonstrate one example for each dataset below.

## LogiQA

One seminar had 18 participants. It is known that :(1) At least 5 young teachers are female; (2) At least 6 female teachers are over middle age; (3) At least 7 young women are teachers; According to the above information, which can be concluded?
Options:
(A) Some young teachers are not women
(B) Some young women are not teachers
(C) There are at least 11 young teachers
(D) There are at least 13 female teachers

## BBH: logical deduction three objects

The following paragraphs each describe a set of three objects arranged in a fixed order. The statements are logically consistent within each paragraph. In a golf tournament, there were three golfers: Ada, Mel, and Mya. Mya finished below Ada. Mel finished above Ada.
Options:
(A) Ada finished last
(B) Mel finished last
(C) Mya finished last

## GSM8k

A 40 meters rope was cut into 2 parts in the ratio of 2:3. How long is the shorter part?

## MATH: Algebra Level 1

$361 + 2(19)(6) + 36 = x$. Solve for $x$.

## ContextHub: Abstract - Level 2

(wqeq or mnze) $\rightarrow$ zkx.
(NOT ttjmx) $\rightarrow$ kottz.
(kottz or zkx) $\rightarrow$ pofk.
Given pofk is False, what is the value of ttjmx?

## LiveBench: reasoning - zebra puzzle

There are 3 people standing in a line numbered 1 through 3 in a left-to-right order.
Each person has a set of attributes: Nationality, Music-Genre, Transport.
The attributes have the following possible values:
- Nationality: spanish, argentine, canadian
- Music-Genre: punk, rock, reggae
- Transport: train, jet-ski, trike
and exactly one person in the line has a given value for an attribute.

Given the following premises about the line of people:
- the person who is argentine avoids getting on a train
- the person who is spanish is somewhere between the person who listens to punk and the person who listens to rock
- the person who listens to punk is not anywhere to the right of the person that travels by trike
- the person who listens to punk is on the immediate right of the person that travels by jet-ski

Answer the following question:
What is the nationality of the person who listens to rock? Return your answer as a single word, in the following format: ***X***, where X is the answer.

### A.3 TRAINING DETAILS

For self-training of `TypedThinker`, we use the splits in the original papers for LogiQA and follow the split of Toshniwal et al. (2024) for GSM8k and MATH. For BBH, we utilize 16 English multiple-choice tasks and randomly select 100 examples per task as the test set, with 20 examples as the hold-out validation set. The detailed statistics are listed in Table 9. Finally, the curated generation dataset covers 67.2% problems on the LogiQA benchmark, 69.7% on BBH, 74.88% on GSM8k, and 36.27% on MATH. We finetune a unified meta-thinker for both math and logical problems, and a unified reasoner for all reasoning types. We use Huggingface (Wolf et al., 2019) with deepspeed (Rasley et al., 2020). The finetuning is conducted on 2 A6000 GPUs. The batch size is 64 and the learning rate is $1e-5$. The maximum epoch is 3 for the meta-thinker and 2 for the reasoner.

### A.4 ANALYSIS OF TYPED REASONING

**Accuracy of Typed Reasoning** We calculate the accuracy for each reasoning type on our empirical dataset $\bar{D}$, shown in Figure 6. We can find that on LogiQA and MATH, the accuracy of different reasoning types is similar. However, deductive and analogical reasoning outperform the other two on BBH while inductive and abductive reasoning are more effective. The results illustrate that after our carefully designed demonstration for each reasoning type, LLM's capabilities in other reasoning types achieve comparable performance with deductive reasoning. This ensures the quality and the balance of our collected dataset on each reasoning type.

Comparing Figure 1 and Figure 6, we can see that if correctly selected, the specific reasoning type can enhance the model performance by handling problems that cannot be solved by other reasoning types, such as inductive on MATH. However, the unsuitable reasoning type can also mislead the model, leading to poor performance.

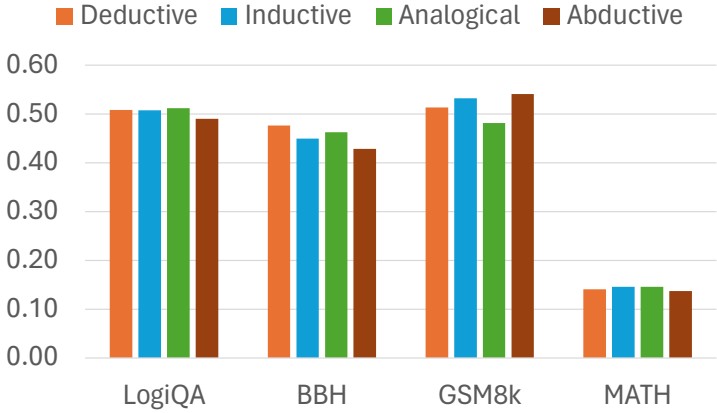

Figure 6: Accuracy of the solutions on different reasoning types. It indicates that the effectiveness of reasoning types varies in different problems.

**Diversity of Typed Reasoning** To further verify whether the reasoning types can make the solutions more diverse, we compare the diversity between solutions under different sampling settings in Table 10. We use Levenshtein Distance (Levenshtein, 1966) and the n-gram overlaps between sentences to evaluate diversity. Specifically, for $K$ generations $G = \{g_1, \cdots, g_K\}$ of the same problem, we calculate the distance between each pair and normalize them with the sentence length. Then the average distance over these paired results is used as the distance of these $K$ generations. If we denote the normalized Levenshtein Distance function as $f_{ld}$, this process can be represented as:

$$f_{ld}(G) = \frac{2}{K(K-1)} \sum_{i=0}^{K} \sum_{j=i+1}^{K} f_{ld}(g_i, g_j). \tag{1}$$

The calculation of the n-gram overlap is defined in the same way. For each setting, we present the average score over the problems in the test set in Table 10. A larger Levenshtein distance and a smaller overlap indicate a more diverse solution set. The zero-shot setting does not include examples

in the prompt, and the zero-shot setting + types only include the definition of the reasoning type (as listed in Table 8). The few-shot setting has 5 examples, and the few-shot setting with types has different 6 examples for each type. For zero-shot / few-shot @5, we use repeated sampling with temperature = 1 for 5 times. For zero-shot / few-shot + 5 types, we sample one solution per reasoning type.

From Table 10, we can see that after adding the reasoning types, the diversity of both zero-shot and few-shot increases significantly. It indicates that the introduction of various reasoning types can make the LLM's reasoning more diverse. We can also find that in most cases, the few-shot with reasoning types has the highest diversity, while in BBH, the zero-shot setting can benefit more from the reasoning types.

Table 10: Adding reasoning types can enhance diversity in both zero-shot and few-shot sampling settings. It can significantly increase the distance and reduce the n-gram overlaps between generations. For each setting, we use Mistral 7B to sample 5 solutions with temperature = 1. @5 indicates repeated sampling 5 times, + 5 types indicates sampling one solution per reasoning type. The diversity is averaged over the whole test set.

| Benchmark | Sampling Setting | Levenshtein Distance ↑ | Unigram overlap ↓ | 4-gram overlap ↓ |
|---|---|---|---|---|
| LogiQA | Zero-shot @ 5 | 0.304 | 0.588 | 0.537 |
| | Zero-shot + 5 types | 0.600 | 0.258 | 0.186 |
| | Few-shot @ 5 | 0.573 | 0.232 | 0.123 |
| | Few-shot + 5 types | **0.644** | **0.175** | **0.077** |
| BBH | Zero-shot @ 5 | 0.517 | 0.310 | 0.216 |
| | Zero-shot + 5 types | **0.712** | **0.128** | **0.050** |
| | Few-shot @ 5 | 0.599 | 0.224 | 0.119 |
| | Few-shot + 5 types | 0.650 | 0.175 | 0.076 |
| GSM8k | Zero-shot @ 5 | 0.624 | 0.195 | 0.091 |
| | Zero-shot + 5 types | 0.683 | 0.151 | 0.054 |
| | Few-shot @ 5 | 0.498 | 0.312 | 0.176 |
| | Few-shot + 5 types | **0.710** | **0.137** | **0.048** |
| MATH | Zero-shot | 0.673 | 0.157 | 0.070 |
| | Zero-shot + 5 types | 0.729 | **0.112** | **0.035** |
| | Few-shot | 0.659 | 0.174 | 0.080 |
| | Few-shot + 5 types | **0.732** | 0.115 | 0.038 |

## A.5 MORE EXPERIMENTAL RESULTS

### A.5.1 RESULTS ON MORE BACKBONE LLMS

We conducted further experiments using Qwen 2-7B-Instruct (Bai et al., 2023) as our backbone LLM. The Qwen series of open-source large language models have demonstrated comparable or even superior performance to the Mistral and LLaMA families across multiple tasks. The results are shown in Table 11. Our method achieves approximately 7% improvement over the few-shot baseline in both single-generation and majority-vote settings (+SC @5). These results demonstrate that `TypedThinker` is a general and effective method for enhancing the reasoning capabilities of various LLMs.

Table 11: Qwen 2-7B-Instruct results. The annotations are the same with Table 1.

| | LogiQA | BBH | GSM8K | MATH | Avg. |
|---|---|---|---|---|---|
| Few-shot | 0.552 | 0.471 | 0.646 | 0.417 | 0.521 |
| + SC @ 5 | 0.579 | 0.554 | 0.763 | 0.497 | 0.598 |
| CoT Selection | 0.554 | 0.516 | 0.772 | 0.451 | 0.573 |
| + SC @ 5 | 0.560 | 0.528 | 0.780 | 0.497 | 0.591 |
| Zeroshot MoR | 0.573 | 0.498 | 0.492 | 0.407 | 0.493 |
| Fewshot MoR | 0.589 | 0.581 | 0.889 | 0.551 | 0.652 |
| `TypedThinker` | 0.595 | 0.534 | 0.779 | 0.474 | 0.596 |
| + SC @ 5 | 0.644 | 0.584 | 0.880 | 0.565 | **0.668** |

Table 12: `TypedThinker` outperforms other baselines on LiveBench without extra finetuning. Here the results are based on the majority vote over 5 responses (+SC @5).

|  | Mistral 7B | LLaMA3 8B |
|---|---|---|
| Few-shot | 0.178 | 0.200 |
| CoT Selection | 0.244 | 0.200 |
| TypedThinker | **0.267** | **0.267** |

### A.5.2 RESULTS ON MORE BENCHMARKS

We further evaluate our methods on LiveBench (White et al., 2024) to test the generalization capability of our method. The experimental settings are the same as described in Section 4.5. The results are shown in Table 12. It demonstrates that `TypedThinker` outperforms other baselines, further supporting its generalization capability across diverse tasks.

### A.5.3 MORE ABLATION STUDIES

The primary reason for comparing our method with the few-shot baseline is that fine-tuning for specific reasoning types is an integral part of our approach. Therefore, we evaluate the impact of our fine-tuned reasoner through a separate ablation study. However, comparing `TypedThinker` to few-shot baselines without fine-tuning may not fully account for the benefits of fine-tuning. Therefore, we conduct two more experiments to verify the influence of the fine-tuned LLMs.

**Comparison with base LLM + one module** Ablation studies in Section 4.4 investigate the contribution of each component by removing one component each time. Here we provide additional ablation results by adding one component to the base LLM each time, resulting in two variants: **Base LLM + Meta-thinker** and **Base LLM + Collection**. For a more reliable conclusion, we ran experiments three times to calculate the average and std and present the result in Table 13 and 14.

Table 13: Mistral 7B results are based on three repetitive experiments. Avg. indicates the average accuracy over four benchmarks.

|  | LogiQA | BBH | GSM8K | MATH | Avg. |
|---|---|---|---|---|---|
| Few-shot | $0.493 \pm 0.007$ | $0.347 \pm 0.01$ | $0.372 \pm 0.014$ | $0.071 \pm 0.003$ | $0.321 \pm 0.006$ |
| CoT Selection | $0.475 \pm 0.009$ | $0.361 \pm 0.01$ | $0.377 \pm 0.011$ | $0.104 \pm 0.008$ | $0.329 \pm 0.004$ |
| LLM + Meta Thinker | $0.512 \pm 0.003$ | $0.377 \pm 0.006$ | $0.379 \pm 0.004$ | $0.106 \pm 0.009$ | $0.343 \pm 0.004$ |
| LLM + Collection | $0.519 \pm 0.007$ | $0.398 \pm 0.005$ | $0.363 \pm 0.004$ | $0.086 \pm 0.008$ | $0.342 \pm 0.001$ |
| TypedThinker | $0.553 \pm 0.004$ | $0.430 \pm 0.008$ | $0.390 \pm 0.012$ | $0.103 \pm 0.01$ | $0.369 \pm 0.006$ |

Table 14: LLaMA 3 8B results are based on three repetitive experiments. Avg. indicates the average accuracy over four benchmarks.

|  | LogiQA | BBH | GSM8K | MATH | Avg. |
|---|---|---|---|---|---|
| Few-shot | $0.569 \pm 0.003$ | $0.319 \pm 0.006$ | $0.476 \pm 0.004$ | $0.102 \pm 0.005$ | $0.366 \pm 0.001$ |
| CoT Selection | $0.558 \pm 0.011$ | $0.376 \pm 0.007$ | $0.360 \pm 0.006$ | $0.104 \pm 0.009$ | $0.349 \pm 0.007$ |
| LLM + Meta Thinker | $0.538 \pm 0.005$ | $0.434 \pm 0.005$ | $0.508 \pm 0.005$ | $0.118 \pm 0.005$ | $0.400 \pm 0.001$ |
| LLM + Collection | $0.574 \pm 0.011$ | $0.497 \pm 0.004$ | $0.438 \pm 0.005$ | $0.109 \pm 0.006$ | $0.404 \pm 0.001$ |
| TypedThinker | $0.546 \pm 0.004$ | $0.534 \pm 0.003$ | $0.535 \pm 0.001$ | $0.203 \pm 0.009$ | $0.455 \pm 0.002$ |

Results show that the retrieval component improves performance on logical tasks but may mislead models on mathematical datasets. This is consistent with our findings in the ablation study in Table 2: the retrieved solutions with digits may mislead the model. Meanwhile, compared with the ICL reasoner, our finetuned reasoner shows better capability in identifying the suitable reasoning type.

### A.6 DISCUSSION ON MORE REASONING PROBLEMS

In this paper, we mainly focus on logical and math reasoning problems. However, our `TypedThinker` can also be extended to symbolic or commonsense reasoning without extra ef-

Table 15: `TypedThinker` performs best on the abstract category of Contexthub. Here the results are based on the majority vote over 5 responses (+SC @5).

|  | Mistral 7B | LLaMA3 8B |
| --- | --- | --- |
| Few-shot | 0.25 | 0.35 |
| CoT Selection | 0.55 | 0.40 |
| Zero-shot MoR | 0.55 | 0.35 |
| Few-shot MoR | 0.45 | 0.45 |
| Self-Discover | 0.25 | **0.55** |
| `TypedThinker` | **0.75** | **0.55** |

forts. For example, the problems under *abstract* category in ContextHub can be viewed as symbolic reasoning, as shown in A.2.2. Therefore, in addition to the overall performance shown in Table 4, we present specific performance on the *abstract* category of symbolic reasoning in Table 15. As we can see `TypedThinker` outperforms baseline methods, even without further fine-tuning the meta-thinker and reasoner for this symbolic reasoning task.

While the LogiQA dataset contains problems requiring commonsense reasoning, the dataset lacks explicit annotations (such as a specific category) for such tasks. For example, here is one case that requires commonsense knowledge about manufacturing costs, market dynamics, and consumer preferences.

---

**LogiQA: A commonsense reasoning example**

Traditionally, the most highly sought cars have been the sports cars and similar two-door models. Nevertheless, Zincstone Motors has chosen to eliminate the last two-door models and produce only four-door models. Which of the following would, if true, most help to explain Zincstone Motors' strategy?
Options:
(A) In almost every instance, Zincstone Motors models lead all comparable models of competitors in fuel efficiency and have lower average maintenance costs as well.
(B) After a spate of recent additional safety requirements, the cost of frame and doors of Zincstone Motors' standard two-door models are now three times as expensive as standard four-door frame and doors.
(C) Many of Zincstone Motors models are exported and sold overseas, including in some countries like Japan, which import a significant number of cars into the United States.
(D) As American consumers lose access to car manufacturers who produce two-door cars, and as two-door cars occupy smaller and smaller shares of the United States car market, American consumers' tastes tend to shift from two-door cars.

---

For reasoning problems that significantly differ from the existing domains (logic and math), additional demonstrations tailored to the task are recommended to guide reasoning. For example, inductive reasoning might involve deriving a general program from exemplar input-output test cases in code generation (Shao et al., 2024; Yang et al., 2024). In such cases, providing a few task-specific examples or conducting light fine-tuning on the reasoner can further enhance performance.

## A.7 IMPACT STATEMENT

This work will enhance current LLMs with better reasoning capability, which can make them more useful in problem-solving. There might be some potential societal consequences of our work, none of which we feel must be specifically highlighted here. However, it might be misused as we release all our code and data for reproduction. We will try our best to avoid the potential misuse.

Table 16: Examples of four reasoning types.

| Type | Query | Example |
|---|---|---|
| **Deduction** | Alice, Bob, and Claire are dancers at a square dance. At the start of a song, they each have a partner: Alice is with Lola, Bob is with Rodrigo, and Claire is with Patrick. Throughout the song, Alice and Bob switch partners; Claire and Bob switch; Finally, Bob and Alice switch. At the end of the dance, Alice is dancing with Options:
(A) Lola (B) Rodrigo (C) Patrick | (0) At the start: Alice: Lola, Bob: Rodrigo, Claire: Patrick.(1) Alice and Bob switch : Alice: Rodrigo, Bob: Lola, Claire: Patrick. (2) Claire and Bob switch : Alice: Rodrigo, Bob: Patrick, Claire: Lola. (3) Bob and Alice switch : Alice: Patrick, Bob: Rodrigo, Claire: Lola. At the end of the dance, Alice is dancing with Patrick. So the answer is $\boxed{(C)}$. |
| **Induction** | Students who told a lie overestimated how many people could detect it. Volleyball players performing poorly thought teammates noticed more than they actually did. A student wearing a funny T-shirt expected everyone to notice, but only a few did. Which option best illustrates the statements above?
(A) People overestimate how often others notice their appearance and behavior.
(B) People rarely notice the appearance or behavior of others.
(C) We are less observant of others' appearance and behavior than we think.
(D) People are less aware of their appearance and behavior than others are." | By examining each of the scenarios mentioned, we notice all these examples indicate that individuals overestimate the level and extent of attention their actions, appearances, or behaviors receive from others. This consistency across different contexts illustrates a broader psychological phenomenon. So the answer is $\boxed{(A)}$. |
| **Abduction** | The integer m is between 30 and 80 and is a multiple of 6. When m is divided by 8, the remainder is 2. Similarly, when m is divided by 5, the remainder is 2. What is the value of m? | To solve this problem using abductive reasoning, we assume that one possible value of m exists that abides by the constraints and check if this assumption holds.1. Fi rst, filter values of m that are multiples of 6 between 30 and 80. 2. Next, apply the condition that when m is divided by 8, the remainder is 2. Only 42, 66 fit this condition.3. Apply the third condition, that when divided by 5, m should leave a remainder of 2. Testing the applicable values so far and find 42 meets the requirement. So the answer is $\boxed{42}$. |
| **Analogy** | John is 24 years younger than his dad. The sum of their ages is 68 years. How many years old is John? | Retrieval: Question: Lisa is 10 years younger than her mom. The sum of their ages is 70 years. How old is Lisa?
Answer: Lisa is 30 years old and her mom is 40 years old.
These are solved using the same approach as the problem about John and his dad's ages, i.e., setting up two equations based on the information given and then solving for the two variables representing the ages. Therefore, for the given question, John is $\boxed{22}$ years old. |

