# OpenReview forum: "TypedThinker: Diversify Large Language Model Reasoning with Typed Thinking"
_ICLR.cc/2025/Conference — ICLR 2025 Poster_

### Official Review · Reviewer_ncpz · 2024-10-28

**Soundness:** 3
**Presentation:** 3
**Contribution:** 2
**Rating:** 6
**Confidence:** 4

**Summary:**

This paper proposed a novel TypedThinker to incorporate four diverse reasoning types and enhance LLMs' problem-solving abilities. The proposed framework first selected appropriate reasoning types for given problems with a fine-tuned meta-thinker module, and then conducted the selected type of reasoning with a fine-tuned reasoner module and retrieved experiences as few-shot examples. The authors conducted extensive experiments and demonstrated the effectiveness and generalization of the proposed method.

**Strengths:**

1. The idea of different reasoning types in problem-solving is novel and inspiring.
2. The authors conducted sufficient data analysis to support the necessity of applying different reasoning types.
3. The authors conducted extensive experiments and in-depth analysis of the proposed method to demonstrate its effectiveness.

**Weaknesses:**

1. Although the idea of the paper is novel, the technical contribution seems to be limited. The authors seem to simply fine-tune a model to select the reasoning type and use it to search the examples and specify the prompt.
2. Some technical details may be misleading. For example, the memory usually refers to data buffers frequently updated in inference, but in this paper it is updated during training and fixed in inference. It would be better to clarify more clearly. Besides, some details used in training and inference are mixed such as the accuracy measurement and memory storage, which makes it somewhat confusing. It would be better to clarify the framework from the input to the output for training and inference.

**Questions:**

See the Weaknesses.

---

> ### Author Response · Authors · 2024-11-21
>
> Thank you for your thoughtful feedback and for acknowledging the novelty of our framework and the thoroughness of our experiments. We will revise the manuscript to clarify the role of the memory during training and inference and better outline the framework flow from input to output, addressing any potential ambiguities. Your comments will help us refine the paper, and we greatly appreciate your insights.
>
> **W1: Although the idea of the paper is novel, the technical contribution seems to be limited. The authors seem to simply fine-tune a model to select the reasoning type and use it to search the examples and specify the prompt.**
>
> A1: Thank you for recognizing the novelty of our paper's idea of introducing reasoning types to diversify thinking processes. As outlined in the Introduction, effectively enhancing reasoning with reasoning types involves two significant challenges: *(1) How to identify the appropriate reasoning type for a general reasoning problem, and (2) How to apply a specific reasoning type in problem-solving.*
>
> To address these challenges, we propose TypedThinker, a simple yet effective framework. TypedThinker uses a meta-thinker to select reasoning types and collect demonstrations for each type. However, the absence of ground-truth human annotations for the reasoning types required by each problem and their corresponding reasoning processes presents a significant hurdle. To overcome this, we curate a fine-tuning dataset by diversifying reasoning experiences with reasoning types and introducing an effectiveness score to estimate the suitability of each type. This provides pseudo-labels for reasoning types and demonstrations without requiring human annotations (Section 3.2).
>
> Thus, our contributions extend beyond introducing typed thinking to **include an effective method for optimizing type selection and typed reasoning without human supervision**. The ablation study (Table 2) demonstrates the importance of each TypedThinker component, while the comparison between prompt-based type selection (CoT Selection) and TypedThinker (Table 1) highlights the critical role of the fine-tuned meta-thinker.
>
> ---
>
> **W2: Some technical details may be misleading. For example, the memory usually refers to data buffers frequently updated in inference, …, It would be better to clarify the framework from the input to the output for training and inference.**
>
> A2: Thank you for this suggestion. To avoid potential confusion, we will revise the term “Memory” to “Collection of Demonstrations” to emphasize that it represents a static collection of examples used to guide the reasoning process for specific reasoning types. We will also clarify the distinction between training and inference in the final paragraph of Section 3.1 in our revised manuscript.
>
> Additionally, we have elaborated on the accuracy measurement in Lines 256–258:
>
> > On logical benchmarks, we compare the response with the ground truth using an exact match. For mathematics benchmarks, we follow Toshniwal et al. (2024) to compute accuracy.
> >
>
> We have detailed the memory size in Table 9 (Empirical Dataset) in the Appendix and included anonymized code as supplementary material in the original submission to ensure reproducibility.

---

> > ### Comment · Reviewer_ncpz · 2024-11-22
> >
> > Thanks for your responses. For W2, I can understand the accuracy measurement and the memory size in the paper, but the confusing part is that the paper mixes the steps used in training/data preparation and inference. For example, in the explicit memory part of section 3.1, I think the solution collection step is performed in training/data preparation, and the experience retrieval is in inference. But there is no clear mark to tell them apart, which makes it somewhat confusing to understand the pipeline. The same is to the accuracy measurement in line 162.

---

> > > ### Author Response · Authors · 2024-11-22
> > >
> > > Thank you for your further explanation and suggestions! We will revise the manuscript to address these points and enhance clarity:
> > >
> > > * The description of the accuracy measurement **in Lines 256–258 will be moved to Line 162**, where accuracy measurement is first introduced.
> > > * In **Section 3.1 (Explicit Collection of Demonstration)**, we will clarify the distinction between training and inference by adding: "*TypedThinker collects a set of demonstrations for each type of reasoning during training and retrieves the most relevant ones during inference.*"
> > > * **At the last paragraph of Section 3.1**, we will add a pipeline summary: *"To conclude, TypedThinker collects demonstrations for each reasoning type from the training set. During inference, it uses the meta-thinker to predict an effectiveness score for each reasoning type and retrieves the most relevant demonstration from the fixed explicit collection corresponding to the selected reasoning type."*
> > >
> > > We hope these revisions will reduce confusion and improve the manuscript's clarity. Thank you for your valuable feedback!

---

> > > > ### Author Response · Authors · 2024-11-25
> > > >
> > > > Dear Reviewer ncpz,
> > > >
> > > > Thanks for your invaluable comments, which have helped us improve the quality of our manuscript. We have revised our manuscript according to these suggestions (**the red part in revised Section 3 and Section 4**).
> > > >
> > > > Do you have any further questions or concerns that prevent you from recommending acceptance of our paper? Please let us know and we are eager to clarify any of your concerns.
> > > >
> > > > With best regards,
> > > >
> > > > Authors

---

> > > > > ### Comment · Reviewer_ncpz · 2024-11-26
> > > > >
> > > > > Thanks for your efforts on responses and revision. I have raised my scores.

---

> > > > > > ### Author Response · Authors · 2024-11-27
> > > > > >
> > > > > > Dear Reviewer ncpz,
> > > > > >
> > > > > > Thanks for your recognition of our rebuttal efforts and kindly updating the score. We are very delighted to hear that all of your significant concerns have been resolved.
> > > > > >
> > > > > > With best regards,
> > > > > >
> > > > > > Authors

---

### Official Review · Reviewer_axGG · 2024-11-01

**Soundness:** 3
**Presentation:** 3
**Contribution:** 3
**Rating:** 8
**Confidence:** 4

**Summary:**

The paper presents TypedThinker, a framework that improves LLMs problem-solving ability by integrating multiple reasoning types—deductive, inductive, abductive, and analogical. The authors do this by using a meta-thinker for selecting appropriate reasoning types and a reasoner for execution. Several experimental results demonstrate accuracy improvements over baseline models across various logical and mathematical benchmarks.

**Strengths:**

- The framework is clearly explained and the paper is easy to follow
- Improved accuracy by leveraging different and multiple types of reasoning allowing the models to approach the problem with a better suited strategy and different cognitive point of view
- The novelty of the proposed method is a good contribution of the paper which may significantly improve LM's reasoning ability for some tasks.

**Weaknesses:**

- Scalability and resource requirement due to the maintenance of the memory buffer could increase computational cost.
- TypedThinker depends on self-training and experience retrieval. How generalizable is the learned experience to other different domains? In sect. 4.5 the authors discuss applicability of typedThinker to ContextHub however this remains a collection of problems for logical reasoning. There is no issue with the proposed framework to be suited only for mathematical and logical reasoning, but then this needs to be specified.

**Questions:**

- Can this method be applied to other types of reasoning (symbolic, commonsense etc)?
- How quickly can this approach adapt to an unseen task? Would one or two examples from new task be enough to achieve reasonable accuracy? It would be great to give some analysis to the adaptability and continual learning ability of the proposed approach.

---

> ### Author Response · Authors · 2024-11-21
>
> Thank you for your positive evaluation and for highlighting the strengths of our framework, including its clarity, novelty, and ability to improve LLM reasoning performance. We appreciate your constructive feedback and hope these concerns are thoroughly addressed in our response below.
>
> **W1: Scalability and resource requirement due to the maintenance of the memory buffer could increase computational cost.**
>
> A1: Thank you for raising this concern. Our memory buffer is constructed during the training phase by collecting successful trajectories and remains fixed during inference. The memory size is approximately 10k instances for the logic domain and 5k for the math domain, which is manageable and does not impose a significant maintenance burden. Furthermore, TypedThinker can be applied to new logic or math tasks without requiring updates to the memory buffer, as demonstrated in Section 4.5. For retrieval, we employ the sentence transformer to calculate similarity, a lightweight process compared to the computational cost of LLM inference. This ensures the framework remains efficient and scalable.
>
> ---
>
> **W2: How generalizable is the learned experience to other different domains? In sect. 4.5 the authors discuss the applicability of typedThinker to ContextHub however this remains a collection of problems for logical reasoning. There is no issue with the proposed framework being suited only for mathematical and logical reasoning, but then this needs to be specified.**
>
> A2: We have discussed this concern in the limitations section. This paper focuses specifically on logical and mathematical reasoning. While we believe that extending TypedThinker to broader tasks, such as reasoning in code or general problem-solving domains, is an interesting and promising avenue, it is beyond the scope of this study. We will ensure this limitation is clearly articulated in the revised manuscript.
>
> ---
>
> **Q1: Can this method be applied to other types of reasoning (symbolic, commonsense, etc)?**
>
> A: Thank you for the question. Our TypedThinker framework can indeed be extended to other reasoning domains, such as symbolic or commonsense reasoning. For example, the propositional logic benchmark ContextHub, used in our generalization tests, includes symbolic reasoning tasks (e.g., the *abstract* category). Below is a simple example:
>
> > (wqeq or mnze) → zkx.
> (NOT ttjmx) → kottz.
> (kottz or zkx) → pofk.
> Given pofk is False, what is the value of ttjmx ?
> >
>
> In addition to the overall performance shown in Table 4, we present specific performance (+ SC @5) on the *abstract*category of symbolic reasoning below. TypedThinker outperforms baseline methods, even without further fine-tuning the meta-thinker and reasoner for this symbolic reasoning task.
>
> |  | Mistral 7B | LLaMA 8B |
> | --- | --- | --- |
> | Few-shot | 0.25 | 0.35 |
> | CoT Selection | 0.55 | 0.40 |
> | Zero-shot MoR | 0.55 | 0.35 |
> | Few-shot MoR | 0.45 | 0.45 |
> | TypedThinker | 0.75 | 0.55 |
>
> For commonsense reasoning, while the LogiQA dataset contains problems requiring commonsense reasoning, the dataset lacks explicit annotations (such as a specific category) for such tasks. For example:
>
> > Traditionally, the most highly sought cars have been the sports cars and similar two-door models. Nevertheless, Zincstone Motors has chosen to eliminate the last two-door models and produce only four-door models. Which of the following would, if true, most help to explain Zincstone Motors' strategy?
> Options:
> (A) In almost every instance, Zincstone Motors models lead all comparable models of competitors in fuel efficiency and have lower average maintenance costs as well.
> (B) After a spate of recent additional safety requirements, the cost of the frame and doors of Zincstone Motors' standard two-door models are now three times as expensive as standard four-door frames and doors.
> (C) Many of Zincstone Motors models are exported and sold overseas, including in some countries like Japan, which imports a significant number of cars into the United States.
> (D) As American consumers lose access to car manufacturers who produce two-door cars, and as two-door cars occupy smaller and smaller shares of the United States car market, American consumers' tastes tend to shift from two-door cars.
> >
>
> This task requires commonsense knowledge about manufacturing costs, market dynamics, and consumer preferences.

---

> > ### Author Response · Authors · 2024-11-21
> >
> > **Q2: How quickly can this approach adapt to an unseen task? Would one or two examples from new task be enough to achieve reasonable accuracy? It would be great to give some analysis to the adaptability and continual learning ability of the proposed approach.**
> >
> > A4:  Thank you for this valuable suggestion. The simplest way to adapt TypedThinker to an unseen task is to use the finetuned meta-thinker for reasoning type identification, along with the existing memory as demonstrations. This approach, as shown in Section 4.5, allows for reasonable performance without additional fine-tuning.
> >
> > If the unseen task significantly differs from the existing domains (logic and math), additional demonstrations tailored to the task are recommended to guide reasoning. For example, in **code generation**, inductive reasoning might involve deriving a general program from exemplar input-output test cases [1]. In such cases, providing a few task-specific examples or conducting light fine-tuning on the reasoner can further enhance performance.
> >
> > [1] Case2Code: Learning Inductive Reasoning with Synthetic Data

---

> > ### Comment · Reviewer_axGG · 2024-11-22
> >
> > I thank the authors for thoroughly answering my questions. I think that adding something about other types of reasoning besides math and logic would improve the manuscript. However, I'm satisfied with your replies and I maintain my current evaluation.

---

> > > ### Author Response · Authors · 2024-11-23
> > >
> > > Thanks for the positive feedback! We will later upload our revised manuscript and include some discussions about the other reasoning types.

---

### Official Review · Reviewer_68w7 · 2024-11-02

**Soundness:** 3
**Presentation:** 3
**Contribution:** 1
**Rating:** 5
**Confidence:** 4

**Summary:**

The paper proposes to solve reasoning by a two-step process. First, select a reasoning type from deductive, inductive, abductive and analogical reasoning. This selector is finetuned. Then historical success cases of the selected reasoning type are retrieved from the memory. These retrieved examples along with the selected reasoning type are used to solve the reasoning problem. The paper conducted experiments on Mistral 7B, LLaMDA 3 8B, GPT-4o and MetaMath, on benchmarks such as BBH, GSM8K, MATH etc.

**Strengths:**

1. The paper is well written and easy to follow.
2. The experiments conducted are very thorough to get a whole picture of the impact of each component.
3. It is clearly motivated that diversity is important for superior reasoning performance in LLMs.

**Weaknesses:**

1. The paper lacks novelty: the idea of introducing different thinking types is not new at all. For example, Meta-Prompting [1] introduced the concept of applying different experts to solve different tasks. Self-Discover [2] proposed to select and compose different reasoning modules to customize the solving of each reasoning task.
2. The fundamental assumption of matching each task into one of the 4 reasoning types is flawed. There are many ways of solving reasoning beyond the 4 reasoning types. In addition, each task could involve several different reasoning types at different steps.
3. The proposed method is unnecessarily complex with a fine-tuned reasoner, a meta thinker and a memory.
4. Despite the cost of fine tuning a reasoner, the gain compared to Few-shot methods is not impressive at all and doesn’t seem to justify the complexity of the pipeline. For example, on Mistral 8B, the average gain on 4 tasks from Few-shot SC@5 is only 3.4% (Table 1). There is almost no gain at all when TypedThinker is applied to GPT-4o (Table 5) and MetaMath (Table 6).
5. It is not fair to compare TypedThinker against few-shot methods, since TypedThinker involves fine tuning. Instead, it should be compared against methods such as OPRO [3].

[1] https://arxiv.org/abs/2401.12954

[2] https://arxiv.org/abs/2402.03620

[3] https://arxiv.org/abs/2309.03409

**Questions:**

Overall, the basic assumption of choosing one reasoning type from 4 is flawed. The fact that there is almost no gain at all for strong models such as GPT-4o reflects the method is not touching the core of how to improve LLM's reasoning ability.

---

> ### Author Response · Authors · 2024-11-21
>
> Thank you for your thoughtful feedback and for acknowledging the strengths of our paper. We appreciate your concerns and would like to address them in detail, as we believe our approach offers valuable contributions to the field.
>
> **W1: The paper lacks novelty: the idea of introducing different thinking types is not new at all. For example, Meta-Prompting introduced the concept of applying different experts to solve different tasks. Self-Discover proposed to select and compose different reasoning modules to customize the solving of each reasoning task.**
>
> A1: You mentioned that our idea of introducing different reasoning types is not novel, pointing out work like Meta-Prompting and Self-Discover. We acknowledge that these works explore similar ideas, such as applying different reasoning modules or experts[1,2,3]. However, we would like to clarify our contribution and how it differs from prior work.
>
> Our key novelty lies in **introducing and formalizing four distinct reasoning types (deductive, inductive, abductive, and analogical)** for **instance-level reasoning** and **how to effectively apply them to diverse reasoning problems**. The challenge we address is not just selecting a reasoning module, but selecting **the most appropriate reasoning type for each individual problem** and then applying it to improve LLM performance. Unlike previous methods, we **explicitly focus on instance-specific reasoning** by utilizing a meta-thinker and fine-tuned reasoners. This allows us to **optimize reasoning types for each task**, which, as shown in our experiments, leads to a more effective reasoning process.
>
> We will add a more detailed comparison to Meta-Prompting and Self-Discover in the related work section to make the distinctions clearer.
>
> To further address the concern on Self-discover, we included Self-Discover as a baseline and present its performance below, following its official implementation [4].
>
> Results show that Self-Discover performs poorly on Mistral 7B and LLaMA 3 8B compared to TypedThinker. This may be due to the difficulty these models face in identifying a reusable high-level reasoning structure for diverse instances, especially in datasets like LogiQA, where reasoning structures are highly varied. Additionally, the generated structures often introduce excessive complexity, confusing the models under zero-shot settings, especially for weaker models like Mistral 7B.
>
> | Mistral 7B | LogiQA | BBH | GSM8K | Math | Average |
> | --- | --- | --- | --- | --- | --- |
> | Self-discover | 0.386 | 0.340 | 0.141 | 0.056 | 0.231 |
> |  + SC @ 5 | 0.476 | 0.391 | 0.208 | 0.082 | 0.289 |
> | TypedThinker | 0.554 | 0.423 | 0.386 | 0.092 | 0.364 |
> |  + SC @ 5 | 0.570 | 0.469 | 0.500 | 0.149 | 0.422 |
>
> | LLaMA 3 8B | LogicQA | BBH | GSM8K | Math | Average |
> | --- | --- | --- | --- | --- | --- |
> | Self-discover | 0.493 | 0.425 | 0.587 | 0.200 | 0.426 |
> | + SC @ 5 | 0.540 | 0.543 | 0.701 | 0.278 | 0.515 |
> | TypedThinker | 0.550 | 0.533 | 0.535 | 0.193 | 0.453 |
> | + SC @ 5 | 0.620 | 0.591 | 0.723 | 0.263 | 0.549 |
>
> [1] Improving Factuality and Reasoning in Language Models through Multiagent Debate
>
> [2] Meta-Prompting: Enhancing Language Models with Task-Agnostic Scaffolding
>
> [3] SELF-DISCOVER: Large Language Models Self-Compose Reasoning Structures
>
> [4] https://github.com/kailashsp/SELF-DISCOVER
>
> ---
>
> **W2: The fundamental assumption of matching each task into one of the 4 reasoning types is flawed. There are many ways of solving reasoning beyond the 4 reasoning types. In addition, each task could involve several different reasoning types at different steps.**
>
> A2: We understand your concern that the assumption of categorizing reasoning into four types may be limiting. However, our focus on **logical reasoning** has been informed by established frameworks in cognitive science and logic [5, 6], where these four types serve as fundamental approaches to problem-solving. **While we agree that reasoning can be diverse and multi-faceted**, our method is designed to **enhance reasoning by structuring it around these four core logical types**, which have proven effective in various domains. For other categorizations, our approach is extendable and could incorporate additional reasoning structures in future work.
>
> Furthermore, as we noted in Section 5, **multi-step reasoning** scenarios are a natural extension of our framework. One approach is to use the same reasoning type throughout, aligning with our current experimental settings. Alternatively, TypedThinker could dynamically identify the most effective reasoning type at each step, using the meta-thinker to guide reasoning type selection and intermediate problem-solving. Our work lays the groundwork for such multi-step investigations, which we believe is a promising future direction.
>
> [5] Logical Reasoning ISBN 10.0534176887
>
> [6] Logical Reasoning and Learning, ISBN 9781441914279

---

> > ### Author Response · Authors · 2024-11-21
> >
> > **W3: The proposed method is unnecessarily complex with a fine-tuned reasoner, a meta thinker and a memory.**
> >
> > A3: We acknowledge that the inclusion of a fine-tuned reasoner, a meta-thinker, and memory might seem complex. However, we argue that this complexity is justified because **each component addresses a specific challenge** in improving reasoning.
> >
> > We conducted an ablation study (Section 4.4, Table 2) to analyze the impact of each component. The results show that the contributions of the meta-thinker, memory, and fine-tuned reasoner vary between domains. For instance, the meta-thinker is critical for math problems, while memory has a greater impact on logical tasks. The fine-tuned reasoner consistently enhances reasoning capabilities across domains, demonstrating that all components are indispensable for identifying and applying suitable reasoning types effectively.
> >
> > Furthermore, as shown in Table 1, the fine-tuned reasoner outperforms prompt-based methods like CoT Selection. This indicates the **fine-tuning is primarily an upfront cost** that results in significant long-term benefits, both in terms of reasoning accuracy and efficiency.
> >
> > ---
> >
> > **W4: Despite the cost of fine tuning a reasoner, the gain compared to Few-shot methods is not impressive at all and doesn’t seem to justify the complexity of the pipeline. For example, on Mistral 8B, the average gain on 4 tasks from Few-shot SC@5 is only 3.4% (Table 1). There is almost no gain at all when TypedThinker is applied to GPT-4o (Table 5) and MetaMath (Table 6).**
> >
> > A4: We appreciate your observation regarding the limited gains with strong models like GPT-4o. We agree that the baseline performance of such models is already high, and incremental improvements may not always be as large as with smaller models. However, our results demonstrate that **even strong models benefit from the application of the right reasoning type, which is identified by a small finetined meta-thinker** (based on Mistral 7B). This underscores the potential of using small models to guide larger ones, enabling efficient knowledge transfer and reasoning improvement across models of varying capacities. For instance, in Table 5, the application of TypedThinker to **MetaMath** still results in a noticeable performance improvement.
> >
> > The complexity of our pipeline primarily lies in the fine-tuning phase. However, we emphasize that **once the meta-thinker is optimized using the fine-tuned dataset, it can generalize to other LLMs without additional fine-tuning**. This significantly reduces the overall complexity.
> >
> > ---
> >
> > **W5: It is not fair to compare TypedThinker against few-shot methods, since TypedThinker involves fine tuning. Instead, it should be compared against methods such as OPRO [3].**
> >
> > A5: Thank you for pointing this out. Our primary focus is on diversifying LLM reasoning through logical types and identifying instance-specific reasoning strategies. Optimizing prompt wording (as done in OPRO) is not the main focus of this work, though such techniques can complement TypedThinker and baseline methods.
> >
> > To address concerns regarding fine-tuning, we provide additional ablation results comparing settings without fine-tuning, such as **Base LLM + Meta-Thinker** and **Base LLM + Memory**. For clarity, we include the several lines from Table 1  for better comparison. Results for Mistral-7B show that the retrieval component improves performance on logical tasks but may mislead models on mathematical datasets. This is consistent with our findings in the ablation study in Table 2: the retrieved solutions with digits may mislead the model. Meanwhile, compared with the ICL reasoner, our finetuned reasoner show better capability in identifying the suitable reasoning type.
> >
> > |  | LogiQA | BBH | GSM8K | Math | Average |
> > | --- | --- | --- | --- | --- | --- |
> > | Fewshot | 0.485 | 0.346 | 0.369 | 0.074 | 0.318 |
> > | CoT Selection | 0.474 | 0.361 | 0.372 | 0.095 | 0.325 |
> > | LLM + Meta-Thinker | 0.509 | 0.373 | 0.383 | 0.105 | 0.342 |
> > | LLM + Memory | 0.513 | 0.394 | 0.367 | 0.087 | 0.340 |
> > | TypedThinker | 0.554 | 0.423 | 0.386 | 0.092 | 0.364 |

---

> > > ### Comment · Reviewer_68w7 · 2024-11-22
> > >
> > > After reading through the authors' responses, I am even more convinced that the results presented are not justifying the cost of the complexity of the method:
> > >
> > > 1. Despite that fine-tuning is an upfront cost, comparing a fine-tuned method like TypedThinker to prompting method is NOT a fair comparison. I hope the authors can get this point.
> > > 2. The comparison of LLM+Meta-Thinker and LLM+Memory to CoT Selection is a more fair one. As reflected by the results in the Table, there is only 1.7% and 1.5% gain compared to CoT Selection. We don't know what is the variance of the evaluations and hence I am not sure it is even statistically significant.
> > > 3. Looking at the comparison more carefully, the 1.7% and 1.5% primarily comes from LogiQA while GSM8K and MATH even suffer losses. My understanding is that the breakdown into 4 reasoning types fits to LogiQA naturally while doesn't generalize to other tasks like MATH. This raises a major concern of the generalization of the proposed approach.
> > > 4. I am not convinced by the argument that the method is generalizable when it only works on smaller models while fail to improve the model performance for strong models like GPT-4o.
> > >
> > > Regarding the novelty of the method, the major claims from the authors are `four reasoning types` and `instance-specific reasoning`. Neither is novel in the sense that they are just natural extensions/expansions of general reasoning modules and task-level reasoning that are around in many papers already.
> > >
> > > Based on the concerns of generalizability and novelty of the proposed method, I will keep my score.

---

> > > > ### Author Response · Authors · 2024-11-24
> > > >
> > > > We appreciate that you read our response very carefully and your informative feedback, which has helped improve our paper. Below please see our response to your concerns.
> > > >
> > > > **Q1: Despite that fine-tuning is an upfront cost, comparing a fine-tuned method like TypedThinker to prompting method is NOT a fair comparison.**
> > > >
> > > > A1: Thank you for pointing out this important concern regarding the fairness of comparisons between fine-tuned methods and prompting-based baselines. We agree that comparing TypedThinker to few-shot baselines without fine-tuning may not fully account for the benefits of fine-tuning.
> > > >
> > > > The primary reason for comparing our method with the few-shot baseline is that fine-tuning for specific reasoning types is an integral part of our approach. Therefore, we evaluate the impact of our fine-tuned reasoner through a separate ablation study.
> > > >
> > > > To address the reviewer’s concern, we have conducted additional experiments to ensure a more balanced comparison:
> > > >
> > > > - **Few-shot and Self-Discover Baselines:** We fine-tuned the backbone LLMs for these baselines using reasoning trajectories with the reasoning type (*Empty*) and filtered out incorrect samples. We used a similar amount of training data as that used for fine-tuning our reasoners.
> > > > - **CoT Selection and MoR Baselines:** We used the same fine-tuned reasoners as those employed in TypedThinker.
> > > >
> > > > The updated results are summarized below. We observe that all baselines improved with fine-tuning. **However, TypedThinker continues to outperform all baselines,** demonstrating the effectiveness of combining the meta-thinker and fine-tuned reasoners. Additionally, the performance gap between zero-shot and few-shot MoR was reduced, as the fine-tuned reasoner better aligned the reasoning process with the selected type. Interestingly, we noted a performance drop in the self-discover baseline for LLaMA 3 8B, likely due to misalignment between the fine-tuned trajectories and the reasoning structure required in self-discover.
> > > >
> > > > |  | Mistral 7B Avg. | LLaMA3 8B Avg. |
> > > > | --- | --- | --- |
> > > > | Few-shot | 0.339 | 0.413 |
> > > > |  + SC @ 5 | 0.390 | 0.435 |
> > > > | CoT Selection | 0.365 | 0.432 |
> > > > |  + SC @ 5 | 0.401 | 0.440 |
> > > > | Self-Discover | 0.263 | 0.420 |
> > > > |  + SC @ 5 | 0.314 | 0.497 |
> > > > | Zero-shot MoR @ 5 | 0.366 | 0.385 |
> > > > | Few-shot MoR @ 5 | 0.384 | 0.414 |
> > > > | TypedThinker | 0.364 | 0.453 |
> > > > |  + SC @ 5 | 0.422 | 0.549 |
> > > >
> > > > ---
> > > >
> > > > **Q2: We don't know what is the variance of the evaluations and hence I am not sure it is even statistically significant.**
> > > >
> > > > A2: Thanks for this comments. To address the concern about the improvement, we further conduct this experiment on LLaMA 3 8B, and run each experiment three times to calculate the average and std. The updated results are provided below, showing that the meta-thinker and memory components consistently improve average performance across both Mistral 7B and LLaMA 3 8B models. These results confirm the robustness and reliability of the improvements introduced by our method.
> > > >
> > > > | Mistral | LogiQA | BBH | GSM8K | Math | Average |
> > > > | --- | --- | --- | --- | --- | --- |
> > > > | Fewshot | 0.493 ± 0.007 | 0.347 ± 0.01 | 0.372 ± 0.014 | 0.071 ± 0.003 | 0.321 ± 0.006 |
> > > > | CoT Selection | 0.475 ± 0.009 | 0.361 ± 0.01 | 0.377 ± 0.011 | 0.104 ± 0.008 | 0.329 ± 0.004 |
> > > > | LLM + Meta-Thinker | 0.512 ± 0.003 | 0.377 ± 0.006 | 0.379 ± 0.004 | 0.106 ± 0.009 | 0.343 ± 0.004 |
> > > > | LLM + Memory | 0.519 ± 0.007 | 0.398 ± 0.005 | 0.363 ± 0.004 | 0.086 ± 0.008 | 0.342 ± 0.001 |
> > > > | TypedThinker | 0.553 ± 0.004 | 0.43 ± 0.008 | 0.39 ± 0.012 | 0.103 ± 0.01 | 0.369 ± 0.006 |
> > > >
> > > > | LLaMA 8B | LogiQA | BBH | GSM8K | Math | Average |
> > > > | --- | --- | --- | --- | --- | --- |
> > > > | Fewshot | 0.569 ± 0.003 | 0.319 ± 0.006 | 0.476 ± 0.004 | 0.102 ± 0.005 | 0.366 ± 0.001 |
> > > > | CoT Selection | 0.558 ± 0.011 | 0.376 ± 0.007 | 0.36 ± 0.006 | 0.104 ± 0.009 | 0.349 ± 0.007 |
> > > > | LLM + Meta-Thinker | 0.538 ± 0.005 | 0.434 ± 0.005 | 0.508 ± 0.005 | 0.118 ± 0.005 | 0.400 ± 0.001 |
> > > > | LLM + Memory | 0.574 ± 0.011 | 0.497 ± 0.004 | 0.438 ± 0.005 | 0.109 ± 0.006 | 0.404 ± 0.001 |
> > > > | TypedThinker | 0.546 ± 0.004 | 0.534 ± 0.003 | 0.535 ± 0.001 | 0.203 ± 0.009 | 0.455 ± 0.002 |

---

> > > > > ### Author Response · Authors · 2024-11-24
> > > > >
> > > > > **Q3: Looking at the comparison more carefully, the 1.7% and 1.5% primarily comes from LogiQA while GSM8K and MATH even suffer losses. My understanding is that the breakdown into 4 reasoning types fits to LogiQA naturally while doesn't generalize to other tasks like MATH. This raises a major concern of the generalization of the proposed approach.**
> > > > >
> > > > > A3. Thank the reviewer for the further insightful comments. We would like to clarify that **the incorporation of reasoning types enhances the math performance**, as evidenced by the comparison between Few-shot and LLM + Meta-Thinker. This aligns with our observation in Figure 1, which shows that many math problems, especially on the MATH benchmark, benefit from reasoning types like inductive reasoning.
> > > > >
> > > > > To illustrate this further, we include a detailed example below, with additional examples in Table 11 of the manuscript and the file `examples_math_logic.py` in the uploaded code.
> > > > >
> > > > > > **Query**: The integer m is between 30 and 80 and is a multiple of 6. When m is divided by 8, the remainder is 2. Similarly, when m is divided by 5, the remainder is 2. What is the value of m?
> > > > > >
> > > > > >
> > > > > > **Answer**: To solve this problem using abductive reasoning, we assume that one possible value of $m$ exists that abides by the constraints and check if this assumption holds.
> > > > > > 1. First, filter values of $m$ that are multiples of $6$ between $30$ and $80$. These are $36, 42, 48, 54, 60, 66, 72, 78$.
> > > > > > 2. Next, apply the condition that when $m$ is divided by $8$, the remainder is $2$. Checking through our list we can find that only $42, 66$ fit this condition.
> > > > > > 3. Apply the third condition, that when divided by $5$, $m$ should leave a remainder of $2$. Testing the applicable values so far and find $42$ get a remainder of $2$ while $66$ get a remainder of $6$.
> > > > > > 4. Finally, converge on $m = 42$ as it fits all conditions.
> > > > > > So the answer is $\\boxed{42}$.
> > > > > >
> > > > >
> > > > > On the other hand, the performance degradation on math benchmarks happens when adding the explicit collection. We also discussed this in our ablation study in Section 4.4 Lines 363-365: The math calculation in the retrieved chain-of-thoughts solutions may mislead the reasoner, due to a similar surface context with different numbers. This is also consistent with the observations in [1,2].
> > > > >
> > > > > The performance degradation on math benchmarks occurs when incorporating the explicit collection. As discussed in our ablation study (Section 4.4, Lines 363–365), retrieved chain-of-thought solutions for math problems may mislead the reasoner due to similar surface contexts but differing numerical details. This issue, also noted in prior work [1,2], highlights the challenges of leveraging retrieved examples for numerical tasks.
> > > > >
> > > > > [1] OpenMathInstruct-1: A 1.8 million math instruction tuning dataset.
> > > > > [2] Skill-Based Few-Shot Selection for In-Context Learning
> > > > >
> > > > > ---
> > > > >
> > > > > **Q4: I am not convinced by the argument that the method is generalizable when it only works on smaller models while fail to improve the model performance for strong models like GPT-4o.**
> > > > >
> > > > > A4: Thank you for your comment. We would like to emphasize that our method is orthogonal to the backbone LLM’s capability, and even strong models like GPT-4o can benefit from typed reasoning. For example, a fine-tuned Mistral 7B meta-thinker can effectively improve GPT-4o's performance by identifying the appropriate reasoning type, particularly in single-generation settings, as demonstrated in the comparison between GPT-4o and TypedThinker w/o SC.
> > > > >
> > > > > To further explore this, we conducted additional experiments using a LLaMA 3 8B meta-thinker with GPT-4o. The results show a larger performance improvement, highlighting that a more powerful meta-thinker can enhance the performance of strong backbone models.
> > > > >
> > > > > Our method provides a promising direction for enhancing reasoning capabilities, and investigating the scaling law of TypedThinker on larger models and more extensive training data is a valuable future direction. We appreciate your feedback, which helps to contextualize the broader potential of our approach.
> > > > >
> > > > > |  | LogiQA | BBH | GSM8k | MATH | Avg |
> > > > > | --- | --- | --- | --- | --- | --- |
> > > > > | GPT-4o | 0.76 | 0.84 | 0.97 | 0.89 | 0.865 |
> > > > > |  + SC @ 5 | 0.80 | 0.85 | 0.98 | 0.90 | 0.883 |
> > > > > | TypedThinker with Mistral 7B Meta-thinker | 0.80 | 0.86 | 0.95 | 0.88 | 0.873 |
> > > > > |  + SC @ 5 | 0.81 | 0.90 | 0.96 | 0.91 | 0.895 |
> > > > > | TypedThinker with LLaMA 3 8B Meta-thinker | 0.82 | 0.85 | 0.98 | 0.92 | 0.893 |
> > > > > |  + SC @ 5 | 0.82 | 0.91 | 0.97 | 0.93 | 0.908 |
> > > > >
> > > > > ---

---

> > > > > > ### Comment · Reviewer_68w7 · 2024-11-24
> > > > > >
> > > > > > The gain from `TypedThinker with LLaMA 3 8B Meta-thinker` is still not very impressive other than LogiQA, which is a task I believe is custom-fitted for the 4 reasoning types. Hence, my concern on generalization of the method remains. I do appreciate the authors' effort to address my concerns and the additional ablations, which indeed improves the quality of the paper. Hence, I will raise my score.

---

> > > > > > > ### Author Response · Authors · 2024-11-25
> > > > > > >
> > > > > > > Thanks a lot for your encouraging feedback! We are very grateful that you went over our response for a second time.
> > > > > > >
> > > > > > > While we agree that these reasoning types may be more effective in logical reasoning problems, we also highlight that other reasoning problems such as math reasoning (as shown in GSM8k and MATH) and symbolic reasoning (as shown in the abstract category of Contexthub in Table 15) can also benefit from this human mental thinking methodology.
> > > > > > >
> > > > > > > Once again, thank you very much for the time dedicated to reviewing our paper!

---

> ### Author Response · Authors · 2024-11-24
>
> **Q5: Regarding the novelty of the method, the major claims from the authors are `four reasoning types` and `instance-specific reasoning`. Neither is novel in the sense that they are just natural extensions/expansions of general reasoning modules and task-level reasoning that are around in many papers already.**
>
> A5. Thank you for your thoughtful comments. We respectfully disagree that our work can be viewed as *“natural extensions/expansions of general reasoning modules and task-level reasoning”.*
>
> First, our focus on **logical reasoning types** stems from well-established cognitive strategies inspired by human mental processes. We systematically analyze their role in diversifying LLM thinking (Figure 1, Section 4.3) and demonstrate their importance for improving reasoning performance.
>
> Second, we highlight the instance-level reasoning identification and application is necessary but non-trivial because:
>
> - **Task-level reason strategy insufficient** for diverse tasks, as seen in the performance of Self-Discover, where task-level reason structure fails for instance-level diversity. Similarly, reasoning type identification via prompting (e.g., CoT Selection) often struggles. To overcome the lack of annotations for reasoning type labels in finetuning, we propose **effectiveness estimation**(Section 3.2).
> - **Apply a specific reasoning type is challenging***,* as shown by the performance differences between zero-shot and few-shot MoR and the ablation studies. To address this, we introduce **typed reasoning collections** and fine-tuning (Section 3.2).
>
> Furthermore, we provide a comprehensive investigation of the benefits of **reasoning type diversity (Section 4.2), analyze each component’s contributions (Sections 4.3 and 4.4), and evaluate generalization capabilities (Section 4.5)**. All these observations highlight the contribution of our TypedThinker.
>
> Unlike prior studies, which focus on enhancing specific reasoning types or benchmarking individual reasoning strategies, our work studies how **instance-specific reasoning types** can improve general logical and mathematical problem-solving. We believe this focus has not been explored in prior research, highlighting the contribution of TypedThinker. Thank you for raising this important point, and we hope this clarifies our contributions.
>
> ---
>
> Please let us know if you have further concerns and we are happy to provide more detailed information in the discussion phase.

---

### Official Review · Reviewer_3EVV · 2024-11-04

**Soundness:** 2
**Presentation:** 3
**Contribution:** 2
**Rating:** 5
**Confidence:** 3

**Summary:**

This work introduces a novel framework, *TypedThinker*, designed to improve the problem-solving abilities of LLMsimplementing that reasoning type effectively. by integrating multiple types of reasoning, including deductive, inductive, abductive, and analogical. It addresses two core challenges: 1) choosing the best reasoning type for a given problem and 2) implementing that reasoning type effectively. Using a meta-thinker to select the reasoning type and a reasoner to execute it, TypedThinker can improve accuracy from baseline: by 3.4% for Mistral 7B and 16.7% for LLaMA3 8B on logical and mathematical tasks.

**Strengths:**

- S1. The approach to incorporate multiple reasoning types (deductive, inductive, abductive, and analogical) into consideration for a meta-thinker for LLM reasoning is novel and seems like a generalizable approach for most LLMs.
- S2. There were clear quantitative gains for this approach.
- S3. TypedThinker’s ability to generalize to unseen benchmarks is a strength particularly in regards to generalization.
- S4. The experiments on the two models were very thorough, with various components of the framework ablated (e.g. w/o meta-thinker, w/o finetuning, w/o memory on 4 benchmarks)

**Weaknesses:**

- W1. Accuracy improvement of 3.4% on Mistral 7B seems rather minimal, though improvements on LLaMA3 was quite notable. Since there were such varied performance improvements, it would have been more beneficial to have more models to convincingly validate the findings on this paper.
- W2. The individual methods in this paper does not seem novel enough, as strategies for the individual types of reasoning were studied in numerous work (e.g. https://arxiv.org/pdf/2309.05660 for inductive reasoning). However, I do believe that the overall framework has strong potential and is novel by ensembling these existing strategies and methods for prediction. Since there are existing methods for each of these types, it would have been very effective to compare these existing methods with this framework. This could be possibly used to measure effectiveness for/against existing methodologies, and would have made this work a lot more relevant and connected with existing work.
- W3. While overall performance gain with a unified meta-thinker is a gain, I wonder you could have compared this with a baseline, e.g. if there is a significant difference with simply prompting the model to simultaneously state and explore all these reasoning type for final prediction. If there is not a significant performance gain with respect to this baseline, this seems like an overly contrived framework for marginal gains. If there is a significant performance gain, this work would certainly be more convincing, particularly with regards to needing to differentiate reasoning type for precise prediction rather than having inappropriate mixtures.
- W4. While it is encouraging to see that there were improvements on a rather new dataset probably outside pretraining, I do not think an experiment on a single dataset on 1 task is evidence enough to claim generalization.

**Questions:**

- Q1. Figure 1 shows the percentage distribution of each reasoning types. How was this annotated? Did the dataset come with these groundtruth labels for type of reasoning required?
- Q2. Have you considered other models? Especially since there are new models that explore reasoning in-depth, e.g. GPT-o1, it would be interesting to see if this results hold for those reasoning optimized models.
- Q3. This was primarily done with logical reasoning and math. Is there a particular reason for picking those domains? I wonder if this strategy would be more effective with reasoning domains with more ambiguity e.g. natural language reasoning. Though, I can also understand that those domains may be difficult to experiment with/validate due to the nature of NL.

---

> ### Author Response · Authors · 2024-11-21
>
> Thank you for your thoughtful review and for recognizing the novelty of our framework, the robustness of our experiments, and TypedThinker’s generalization capabilities. *Regarding the concern about limited model validation (W1), we have conducted additional experiments with Qwen 2-7B-Instruct, a model comparable to Mistral 7B and LLaMA3, and found consistent performance improvements. For the request to compare with existing methods and other baselines (W2 and W3), we provide a more detailed analysis. Lastly, addressing other reasoning datasets (W4), we evaluate other methods on a new benchmark Livebench with 3 different reasoning tasks.* We hope these clarifications and updates address your key concerns.
>
> **W1: Accuracy improvement of 3.4% on Mistral 7B seems rather minimal, though improvements on LLaMA3 were quite notable. It would have been more beneficial to have more models to validate the findings on this paper convincingly.**
>
> A1: Thank you for your valuable feedback. In response, we conducted further experiments using Qwen 2-7B-Instruct as our backbone LLM. The Qwen series of open-source large language models have demonstrated comparable or even superior performance to the Mistral and LLaMA families across multiple tasks [1]. The results are summarized below:
>
> |  | LogiQA | BBH | GSM8K | MATH | Average |
> | --- | --- | --- | --- | --- | --- |
> | Few-shot | 0.552 | 0.471 | 0.643 | 0.417 | 0.521 |
> |     + SC @5 | 0.579 | 0.554 | 0.752 | 0.497 | 0.596 |
> | CoT Selection | 0.528 | 0.462 | 0.449 | 0.314 | 0.438 |
> |     + SC @5 | 0.560 | 0.528 | 0.767 | 0.490 | 0.586 |
> | Fewshot MoR | 0.607 | 0.568 | 0.798 | 0.490 | 0.616 |
> | TypedThinker | 0.595 | 0.534 | 0.776 | 0.474 | 0.595 |
> |     + SC @5 | 0.644 | 0.584 | 0.875 | 0.565 | 0.667 |
>
> As shown, our method achieves approximately 7% improvement over the few-shot baseline in both single-generation and majority-vote settings (+SC @5). These results demonstrate that TypedThinker is a general and effective method for enhancing the reasoning capabilities of various LLMs.
>
> [1] Qwen Technical Report
>
> ---
>
> **W2: It would have been very effective to compare these existing methods with this framework. This could be possibly used to measure effectiveness for/against existing methodologies and would have made this work a lot more relevant and connected with existing work.**
>
> A2: Thank you for this insightful suggestion. In our revised manuscript, we will expand the discussion on relevant work (in Related Work). Existing studies often focus on using a single reasoning type to address specific problem categories, which limits their generalizability to broader reasoning tasks [2, 3, 4].
>
> For instance, Hypothesis Search [2] is designed for tasks solvable by a single program, such as ARC and SyGuS. While this approach generalizes well for these tasks, it is infeasible for general reasoning challenges, like logical reasoning and math problems, which cannot be solved with a single universal program.
>
> In contrast, our work emphasizes improving LLMs’ ability to identify and apply diverse reasoning types to solve a wider range of logical and mathematical problems. This contribution distinguishes our approach from existing methods.
>
> [2] Hypothesis Search: Inductive Reasoning with Language Models  https://arxiv.org/pdf/2309.05660
>
> [3] Case2Code: Learning Inductive Reasoning with Synthetic Data
>
> [4] UNcommonsense Reasoning: Abductive Reasoning about Uncommon Situations
>
> ---
>
> **W3: Add baselines for simply prompting the model to simultaneously state and explore all these reasoning type for final prediction.**
>
> A3: Thank you for the suggestion. As described in Section 4.1, we have already included two baselines to evaluate the influence of reasoning types:
>
> 1. **CoT Selection**: A two-step approach where the LLM first selects the most appropriate reasoning type and then applies it to solve the problem.
> 2. **Mixture-of-Reasoning (MoR)**: The LLM applies all five reasoning types (including an empty type) and uses majority voting to determine the final answer.
>
> These baselines can be considered enhanced versions of “*simultaneously stating and exploring all reasoning types.*”
>
> The results, summarized in Table 1 (also shown below), reveal that prompt-based approaches struggle to identify suitable reasoning types, while a simple mixture of all reasoning types does not significantly improve reasoning performance.
>
> |  | Mistral 7B Avg. | LLaMA3 8B Avg. |
> | --- | --- | --- |
> | CoT Selection | 0.325 | 0.342 |
> | Fewshot MoR | 0.388 | 0.400 |
> | TypedThinker | 0.422 | 0.549 |
>
> These comparisons highlight the advantages of our TypedThinker framework.

---

> > ### Author Response · Authors · 2024-11-21
> >
> > **W4: an experiment on a single dataset on 1 task is evidence enough to claim generalization.**
> >
> > A4: Thank you for raising this concern. To address this, we evaluated our approach on LiveBench [5], a benchmark with 18 diverse tasks across 6 categories, specifically designed to minimize data contamination. We used all three tasks (*spatial*, *web_of_lies_v2*, *zebra_puzzle*) from the *reasoning* category, splitting them 70/30 for training and testing.
> >
> > The results, shown below, demonstrate that TypedThinker outperforms other baselines, further supporting its generalization capability across diverse tasks.
> >
> > |  | Mistral 7B | LLaMA 3 8B |
> > | --- | --- | --- |
> > | Few-shot | 0.178 | 0.200 |
> > | CoT Selection | 0.244 | 0.200 |
> > | TypedThinker | 0.267 | 0.267 |
> >
> > [5] LiveBench: A Challenging, Contamination-Free LLM Benchmark
> >
> > ---
> >
> > **Q1: Figure 1 shows the percentage distribution of each reasoning types. How was this annotated? Did the dataset come with these ground-truth labels for type of reasoning required?**
> >
> > A1: Thank you for your question. The experimental setup for Figure 1 is described in Lines 56–60 and detailed further in Section 4.2. The original datasets do not provide ground-truth reasoning labels. Instead, we estimate the effectiveness of each reasoning type through sampling. Specifically, for each problem, we sample 10 solutions for each reasoning type using the Mistral 7B instruct model. A reasoning type is deemed effective if at least one correct solution is found among the 10 samples. While a problem may have multiple effective reasoning types, Figure 1 focuses only on problems associated with a single reasoning type.
> >
> > ---
> >
> > **Q2: Have you considered other models? Especially since there are new models that explore reasoning in-depth, e.g. GPT-o1**
> >
> > A2: Thank you for the suggestion. Our method is orthogonal to the choice of the base LLM and focuses on enhancing broader and more diverse reasoning capabilities rather than specializing in deeper reasoning. As shown in Tables 5 and 6, even advanced models like GPT-4o and MetaMath benefit from the application of TypedThinker.
> >
> > We also experimented with o1-mini as the backbone LLM but observed that its few-shot performance was significantly worse than its zero-shot performance. This is likely because the provided few-shot examples lacked the reflective, iterative reasoning that o1 and o1-mini models inherently excel at, as learned through reinforcement learning [6, 7]. This may explain why OpenAI reports o1 results primarily in a zero-shot setting. Consequently, o1 models are less suitable for our framework, which relies on few-shot prompts.
> >
> > [6] [https://openai.com/index/openai-o1-mini-advancing-cost-efficient-reasoning](https://openai.com/index/openai-o1-mini-advancing-cost-efficient-reasoning/#limitations-and-whats-next)
> >
> > [7] https://openai.com/index/learning-to-reason-with-llms/
> >
> > ---
> >
> > **Q3: I wonder if this strategy would be more effective with reasoning domains with more ambiguity e.g. natural language reasoning.**
> >
> > A3: Thank you for your insightful observation. The LogiQA dataset we used is based on natural language rather than formal logic. It was curated from exams designed to test human critical thinking and problem-solving skills. Similarly, the 16 tasks we selected from the BBH benchmark are also natural language tasks (as described in Lines 830–833), requiring strong natural language understanding to resolve ambiguities and perform reasoning.
> >
> > For clarity, we provide a simplified example below (Table 7 and Figure 11 contain additional examples):
> >
> > > One seminar had 18 participants. It is known that :(1) At least 5 young teachers are female; (2) At least 6 female teachers are over middle age; (3) At least 7 young women are teachers; According to the above information, which can be concluded?
> > Options:
> > (A) Some young teachers are not women.
> > (B) Some young women are not teachers.
> > (C) There are at least 11 young teachers.
> > (D) There are at least 13 female teachers.
> > >

---

> > > ### Author Response · Authors · 2024-11-25
> > >
> > > Dear Reviewer 3EVV,
> > >
> > > Thank you for your thoughtful and detailed review of our manuscript. We greatly appreciate the time and effort you have invested in providing valuable feedback. We have carefully addressed your comments through substantial revisions and hope these changes meet your expectations.
> > >
> > > We welcome any additional feedback you may have and would be grateful for the opportunity to further improve our manuscript based on your insights.
> > >
> > > Authors

---

> > > > ### Comment · Reviewer_3EVV · 2024-12-01
> > > >
> > > > Thank you authors, for your thoughtful responses and patience! I thank the authors for going through each of my concerns to address them carefully, and sometimes even with additional results in such a short period of time. I believe that my concerns regarding the performance inconsistence is well addressed, with additional experiments on Qwen. It indeed shows that TypedThinker is an adequate method that can generalize well across models, despite some limited improvements on Mistral. I also would like to thank the authors for providing additional comments on the experiments themselves, expanding upon their rationale for particular designs. Overall, those comments were helpful.
> > > >
> > > > However, I think I still think that W3 on novelty holds. While I agree that the experimental design and various probing proves that the gains are significant enough, I fail to see how this method is significantly novel. I understand that the authors consider 2 baselines, but my main concern regarding novelty for this paper stands. I maintain my rating as a result, and I thank the authors for engaging with concerted effort and expanded details.

---

> > > > > ### Author Response · Authors · 2024-12-02
> > > > >
> > > > > Thanks for your time to read our response carefully and provide more informative feedback. We are grateful that our experiments and explanations have addressed some of your concerns.
> > > > >
> > > > > We would like to highlight further our novelty and contribution in **introducing and formalizing four distinct reasoning types (deductive, inductive, abductive, and analogical) for instance-level reasoning and how to apply them to diverse reasoning problems effectively.** We appreciate that the reviewer also agrees that *"the ensembling of these reasoning types has strong potential"*. Here we also want to explain why our work on the instance-level reasoning type identification and application is important and non-trivial:
> > > > >
> > > > > - Task-level reason strategies are insufficient for tasks with high diversity. The task-level reasoning structure **Self-Discover** fails for instance-level diversity.
> > > > > - Prompt-based reasoning type identification or simple mixture performs poorly, as shown in **CoT Selection and Mixture-of-Reasoning (MoR)** of response A3.
> > > > > - Applying a specific reasoning type is challenging, as shown by the performance differences between **zero-shot and few-shot MoR and the ablation studies**.
> > > > >
> > > > > To address these challenges, we
> > > > >
> > > > > - propose effectiveness estimation to overcome the lack of annotations for reasoning type labels in finetuning (Section 3.2).
> > > > > - introduce typed reasoning collections and fine-tuning to enhance LLMs’ typed reasoning (Section 3.2).
> > > > > - provide a comprehensive investigation of the benefits of reasoning type diversity (Section 4.2), analyze each component’s contributions (Sections 4.3 and 4.4), and evaluate generalization capabilities (Section 4.5).
> > > > >
> > > > > All these observations highlight the contribution of our TypedThinker.
> > > > >
> > > > > We thank the reviewer again for the invaluable comments, which have helped us improve the quality of the manuscript. With the discussion phase coming to an end, we are also eager to answer any further questions or concerns.

---

### Author Response · Authors · 2024-11-25
**Thanks again and we are looking forward to your continued feedback**

Thank all reviewers very much again for the insightful comments, which have helped us improve the quality of our manuscript. We have been dedicated to absorbing them and providing our responses accordingly.

We summarized our response and the modification of our manuscript ***(the red part of our newly uploaded manuscript)*** into the following categories:

**Updated presentation**

1. In revised Section 2, we added the discussion about the multi-agent / multi-expert framework and the discussion about a specific reasoning type. (**3EVV, 68w7**)
2. In revised Section 3, we clarified the difference between training and inference and changed the “Memory” to “collection of demonstration” to avoid potential confusion. (**ncpz**)
3. In revised Appendix A.2.2, we have provided examples for each dataset to demonstrate different kinds of problems our method can solve  (**3EVV, axGG**)
4. In revised Appendix A.6, we have discussed the other types of reasoning, such as commonsense reasoning and symbolic reasoning. (**axGG**)

**Newly conducted experiments**

1. In revised Appendix A.5.1 (Table 11), we conducted additional experiments on another backbone LLM Qwen-2-7B-instruct as suggested (**3EVV)** and the result is consistent with our original claim.
2. In revised Appendix A.5.2 (Table 12), we conducted additional experiments on one more benchmark Livebench to verify the generalization capability of our method (**3EVV**)
3. In revised Table 2, we added one more baseline Self-Discover and found that our Typedthinker can also outperform it (**68w7**)
4. In revised Appendix A.5.3 (Table 13 and 14), we added more ablation studies to investigate the influence of finetuning (**68w7**)
5. In revised Appendix A.6 (Table 15), we added the performance on the abstract category of Contexthub, which demonstrates the performance on symbolic reasoning. (**axGG**)

---

### Comment · Area_Chair_cTR6 · 2024-11-25
**Reviewer Response**

Dear Reviewers,

The rebuttal discussion period is coming to a close and the paper currently has a mix of positive and negative reviewers. The authors have spent a lot of time responding to each concern -- can you take a look at the author responses and let them know any remaining concerns you have?

Best,
AC

---

### Meta-Review · Area_Chair_cTR6 · 2024-12-23

**Metareview:**

This paper introduces a two-step process of LLM reasoning, by first selecting a reasoning type from deductive, inductive, abductive and analogical reasoning using a fine-tuned selector (“meta-thinker”), and then retrieving related instances (along with the selected reasoning type) to solve the reasoning problem. The paper is well-written and presents extensive empirical results on multiple tasks with multiple LLMs. The proposed two-step reasoning idea is intuitive. The main concerns of this work stem from the assumption that each reasoning problem must belong to one of the reasoning types, This assumption could be wrong in real problems that could involve multiple types of reasoning. Empirical improvements over certain baselines seem also not super impressive.

**Additional Comments On Reviewer Discussion:**

The discussion seems effective as some of the reviewers were convinced to raise their scores. In particular, the authors responded to each of reviewers' comments in details and presented extensive new empirical results, which was appreciated. On the other hand, the main concerns of this paper were not fully resolved: (1) assigning each reasoning problem to one reasoning type seems like a simplistic assumption, (2) how general/useful this proposed approach is when applying to stronger base models on more complex problems is unclear. However, overall this paper presents an interesting investigation of a two-stage reasoning process. which, though simplistic in practice, could still be inspiring to some extent. Their empirical results are also ample.

---

### Decision · Program_Chairs · 2025-01-22

Accept (Poster)